# Left-dominance for resting-state temporal low-gamma power in children with impaired word-decoding and without comorbid ADHD

**Oliver H. M. Lasnick** [1]*, **Roeland Hancock**[2], **Fumiko Hoeft**[1]

**1** Department of Psychological Sciences, University of Connecticut, Storrs, Connecticut, United States of America, **2** Wu Tsai Institute, Yale University, New Haven, Connecticut, United States of America

* oliver.lasnick@gmail.com

## Abstract

One theory of the origins of reading disorders (i.e., dyslexia) is a language network which cannot effectively 'entrain' to speech, with cascading effects on the development of phonological skills. Low-gamma (low-γ, 30–45 Hz) neural activity, particularly in the left hemisphere, is thought to correspond to tracking at phonemic rates in speech. The main goals of the current study were to investigate temporal low-γ band-power during rest in a sample of children and adolescents with and without reading disorder (RD). Using a Bayesian statistical approach to analyze the power spectral density of EEG data, we examined whether (1) resting-state temporal low-γ power was attenuated in the left temporal region in RD; (2) low-γ power covaried with individual reading performance; (3) low-γ temporal lateralization was atypical in RD. Contrary to our expectations, results did not support the hypothesized effects of RD status and poor decoding ability on left hemisphere low-γ power or lateralization: post-hoc tests revealed that the lack of atypicality in the RD group was not due to the inclusion of those with comorbid attentional deficits. However, post-hoc tests also revealed a specific left-dominance for low-γ rhythms in children with reading deficits only, when participants with comorbid attentional deficits were excluded. We also observed an inverse relationship between decoding and left-lateralization in the controls, such that those with better decoding skills were less likely to show left-lateralization. We discuss these unexpected findings in the context of prior theoretical frameworks on temporal sampling. These results may reflect the importance of real-time language processing to evoke gamma rhythms in the phonemic range during childhood and adolescence.

## Introduction

Developmental dyslexia (hereafter referred to as decoding-based reading disorder, or RD) is one of the most common and highly studied specific learning disabilities. Its prevalence is estimated to be 5–10%, although the most common estimates are based largely on data from English-speaking countries [1, 2]. Individuals with RD have profound difficulty with reading despite often having normal-range intelligence (IQ) and a lack of other explanatory variables,

**Data Availability Statement:** Information and/or access to all phenotypic data files and EEG data files are available from the Child Mind Institute Healthy Brain Network (CMI HBN) database, at the

following link: http://fcon_1000.projects.nitrc.org/indi/cmi_healthy_brain_network/. DOI: https://doi.org/10.1038/sdata.2017.181 The provided link takes you to the CMI HBN homepage. From there, a user can directly access and download EEG data ("Data" -> "Access" -> "Neuroimaging"). Upon selecting a data release number, they will be redirected to the NITRC portal. From there, they can create a new NITRC account if necessary. Upon logging in to NITRC after they are redirected from the CMI HBN website, they will be brought to the EEG download page. The download page contains more detailed instructions for batch downloads via AWS or Cyberduck, or direct downloads via the portal. To access full phenotypic data, including behavioral assessments, users first complete a Data Usage Agreement and then create a COINS or LORIS account to download the data as described on the CMI HBN data portal ("Data" -> "Access" -> "Phenotypic"). For this study, data was collected from all available EEG/phenotypic releases at the time of access and filtered according to the inclusion criteria described in the Methods section. Interested researchers may replicate the reported results by downloading this data directly from CMI and then following the instructions outlined in the Methods. The authors did not receive any special access privileges.

**Funding:** Author OHML was supported by the following NIH grant(s): T32DC017703 and F31HD107944; and the NSF grant NRT-UtB1735225. Author RH was supported by the following NIH grant(s): R01HD094834. Author FH was supported by the following NIH grant(s): R01HD094834, R01HD096261, and U24AT011281. This study's funders had no role in study design, data collection and analysis, decision to publish, or preparation of the manuscript.

**Competing interests:** The authors have declared that no competing interests exist.

such as impaired vision [3, 4]. RD is often comorbid with other specific learning disorders, attention deficit hyperactivity disorder (ADHD), language disorders, and anxiety/depression [5–7]. Cognitive profiles for RD often include deficits in letter-sound knowledge, rapid automatized naming (RAN), and phonological awareness (PA), with PA being the strongest predictor of reading outcomes for English-speaking populations [1, 8]. RAN refers to the speed with which one can name a series of presented stimuli, while PA is the ability to explicitly engage one's knowledge of the component speech sounds (phonemes) which make up a word [9].

One neurobiological theory of dyslexia suggests that low-gamma (low-γ) neural activity in the 30–45 Hz frequency range may index speech encoding at the phoneme level. Phonemic processing has been associated with entrainment in the low-γ range when listening to speech [10]. Low-γ phase-locking to stimuli is also modulated by phonological contrast, and shows a differential response in typical and poor adolescent readers during sentence-listening [11].

A number of studies have also reported that lateralized intrinsic activity may be related to effective speech processing, phonological processing, and to RD. Resting-state studies in adults have shown that lateralization of low-range gamma band-power in the superior temporal cortices (left hemisphere [LH] > right hemisphere [RH]) is associated with functional asymmetries in speech processing [12]. Intrinsic activity has been shown to predict lateralized language network activity in primary auditory regions [13]. Other studies have reported atypical functional lateralization in individuals with RD, often with weaker relative LH dominance for low-γ activity [14–16]. A recent review proposed that phonemic processing could also be affected by the right frontoparietal attention network, which in turn exerts downstream effects on the LH dorsal reading network [17]. Lateralized low-γ oscillations during resting-state have also been associated with the ability to perceive speech in noise in young children [18].

However, the observed low-γ range in the left superior temporal cortex and its left-right asymmetry have primarily been investigated in task-based contexts, such as sentence-listening [11] and auditory steady-state response paradigms [14]. Whether similar and robust trends exist in resting-state EEG is still unknown. To this point, intrinsic lateralization of resting-state EEG power in the low-γ range in the superior temporal cortices has been reported. In [13], the authors did report that for their sample of adults there is (1) a correlation between fMRI activity and neural oscillations during resting-state at syllabic and phonemic rates; and (2) resting-state lateralization for low-γ activity in the primary auditory cortex; but the latter effect did not extend to the posterior superior temporal cortex. These effects were also not found in another study on adults, which reported only a non-significant trend for left-lateralization in the 28–40 Hz range at rest [12]. Results therefore remain inconclusive.

The literature on low-γ band-power during rest in children specifically, and its relationship to phonological processing and reading, is sparse. Gamma activity changes with development, with induced (non-phase-locked) amplitudes at 20 and 40 Hz decreasing from age 8 to late adolescence (~19 years); synaptic pruning may be a driving force behind these changes [19]. Thompson et al. [18] did find resting-state left-lateralization of low-γ rhythms in children aged 3–5 years that covaried with speech-in-noise perception skills. They suggested endogenous (resting-state) hemispheric specialization for low-γ oscillations may support speech processing in challenging listening environments during early childhood. Two other papers reported links between resting frontal low-γ power and early cognitive and language skills (including nonword reading) in young pre-readers [20, 21]. These three studies [18, 20, 21] sampled pre-readers aged 3–5 and did not look at effects in the superior temporal cortices. They also did not examine relationships with phonological processing, reading and RD risk. There is one study to our knowledge that examined resting-state EEG at the low-γ band in RD children: the authors compared cortical sources of eyes-closed resting state EEG at various frequency bands between a small sample of typical readers (TRs) and children with RD [22]. Interestingly, they

reported null findings for all bands (including delta, theta, and gamma) except alpha rhythms. Earlier resting-state studies in RD children only examined group differences in the theta and alpha rhythms [23]. Therefore, a more conclusive investigation as to whether there is reduced band-power and atypical lateralization in the temporal lobes that is related to phonological processing, reading, and RD in children is warranted.

The aim of the current study is to investigate low-γ power and lateralization at rest in children with and without RD in an age range where children are learning to read (6 to 13 years), expanding prior resting-state studies in beginning readers and adolescents. Prior results have shown that abnormal low-γ phase-locking has been linked to deficits in phonological processing [11]; that those with RD show altered lateralization patterns [15, 16, 24]; that reduced left-lateralized entrainment in the low-γ range is correlated with poorer phonological processing and rapid naming in RD [14]; and that there are developmental effects on low-γ power during typical development [19]. What is lacking in the literature is a clear consensus on resting-state dynamics in the developing reading and language network. Such a consensus allows for a better understanding of the development of children's language and reading processes. Endogenous low-γ power, if associated with phonological processing, reading, and RD, would indicate both a potential biomarker and a more generalized processing deficit in the intrinsic auditory and language network in RD. Such a biomarker would suggest that the functional organization of the 'RD brain' diverges relatively early.

### Hypotheses

Analyses were designed to demonstrate whether temporal gamma power is attenuated within the LH in the RD population, reflects reduced left-lateralization compared to TRs, and is predicted by phonemic decoding scores. If gamma power is attenuated in the LH temporal region of RDs, and/or the typical left-lateralization pattern is reduced/reversed compared to TRs, this would suggest that there is inherent functional organization to the brain which localizes processing of hierarchically-structured timescales for auditory stimuli to a dominant hemisphere; and that this hierarchical processing is atypical at phonemic timescales in children with decoding deficits. Additionally, we aim to determine whether any such differences are associated with individual performance in reading-related skills, specifically pseudoword (fake but pronounceable 'words') decoding. We predict a positive relationship between low-γ band-power and individual differences in phonemic decoding; and that the effect will be stronger in the left hemisphere compared to the right (LH > RH). This is based on prior work in adults which showed that the LH is more tuned for the processing of gamma band information >20 Hz, while slower delta and theta bands dominate in the right (for a review of oscillation and lateralization differences in RD, see [17]).

## Materials and methods

### Participants

All data came from the Child Mind Institute Healthy Brain Network Project (CMI HBN), an online repository of multi-site imaging and behavioral data [25]. Neuroimaging and EEG data is publicly available for download on the CMI HBN website (see Data Availability statement). Behavioral and phenotypic data are available to researchers upon completion of a Data Usage Agreement, to be signed by an authorized institutional official different from the principal investigator. They are provided in a de-identified/anonymized format such that none of the authors have access to any information which could be used to re-identify the participants. Therefore, additional participant consent or involvement was not required, and no further

approval was sought from the University of Connecticut's Institutional Review Board. Data were accessed for research purposes in May 2020.

The CMI dataset participants range in age from 5–21 years old [25], however the age range for this study was restricted to those aged 6–13 years. Reasoning was threefold [8, 26]: (1) age 6 is when most children achieve early reading skills such that both reading metrics and individual differences are stable and reliable (standardized scores on tests of word reading may be derived starting at 6, and a majority of children with reading impairments are identified during this time frame), (2) within this age range, PA contributes to individual differences in word-reading and word-reading is reflective of decoding ability, and (3) developmental changes in the brain possibly related to neural oscillations in the gamma range have been documented during this time frame [19].

Other inclusion criteria were a full-scale IQ [Wechsler Intelligence Scale for Children (WISC), internal consistency of primary index scores = .88-.93] score >70 (to ensure that we had a representative sample population with a normal distribution, that the participants were cognitively able to complete the assessments reliably, and that reading deficits observed in our RD sample were not attributable to intellectual disability) [27] and completion of the Test of Word Reading Efficiency (TOWRE-2, alternate-form reliability = .91-.95), which consists of the phonemic decoding (PDE) and sight-word efficiency (SWE) subtests [28, 29]. TOWRE is a timed word reading measure wherein the participant is presented with a list of either words (SWE) or pseudowords (PDE) and instructed to read each aloud, in order, as quickly and accurately as possible. They are given 45 seconds to do so. Higher scores reflect a larger number of words read and/or pronounced correctly. These two subtests create an overall reading efficiency composite score, which is frequently used as RD classification criteria in research; total scores that fall below 85 (with the expected population average being 100 with ±15 equivalent to 1 SD) are often used to indicate poor reading (for recent examples, see [30–32]).

For the group-based analyses, two main groups were generated: (1) an RD group with both PDE and SWE scores <85, to ensure that the participants had both phonemic decoding and (non-compensated) word-reading deficits; and (2) a TR group who (a) had not received a prior clinician diagnosis of RD and (b) had both TOWRE subtest scores >90. Because the practice of excluding comorbidities or placing restrictions on participant samples (e.g., only including right-handed males) inherently reduces the generalizability of findings—especially in relation to RD, for which comorbidities are highly prevalent and may affect developmental outcomes [33]—we allowed our sample to have certain clinical diagnoses that often co-occur with reading deficits at elevated rates compared to the general population. We expected that rates of all included diagnoses would be higher in those with RD compared to the TRs. These included depression, anxiety, attention-deficit hyperactivity disorder (ADHD), and other specific learning disorders such as dyscalculia, dysgraphia, and language disorder. Identification of these comorbidities was based on consensus diagnoses, following the procedure used by CMI for the HBN project. The consensus diagnoses are based on a combination of (1) psychiatric and medical history collected during a prescreening phone interview (a participant may have previously received an official diagnosis prior to involvement in the project); (2) clinician-administered diagnostic assessments and observations, including administration of the Kiddie Schedule for Affective Disorders and Schizophrenia (KSADS, Cronbach's α = .66, range: .25-.86) [34], which can be used to identify ADHD, major depressive disorder, generalized anxiety disorder, conduct disorders, and schizophrenia, among others; and (3) clinician-administered follow-up assessments, such as the Clinical Evaluation of Language Fundamentals (CELF-5, internal consistency of index scores = .92-.97) for identifying suspected language disorder [35], as well as other language-centric assessments such as TOWRE-2. For a full set of all assessments, see [25]. Official diagnoses are based on clinician assessment of a combination

of results from the relevant tests and self-reported information from children, parents, and teachers.

All comorbidities were permitted to be included, with the exception of severe intellectual disability due to our IQ cutoff criteria. Comorbid diagnosis of ADHD was also restricted to the RD population on account of its common association with RD [6]. Children with low-average TOWRE scores (85–90) were not included, as we wanted to ensure the presence of a gap in reading ability between the two main groups. After applying our exclusion criteria, some comorbidities were not represented in our sample (for instance, conduct disorders; see Results). We acknowledge that this decision constitutes a tradeoff between perfectly matched TR and RD samples and a more naturalistic approach that allows for greater neurodiversity in our sample. This could result in group differences being observed that are driven at least in part by the effects of elevated comorbidities in RD, particularly ADHD; on the other hand, allowing other comorbidities in both groups could also artificially 'deflate' true group differences. Demographic differences between groups will be controlled for by including them as confounding variables in our statistical analyses.

This gave a sample size of N = 338 (TRs: N = 192; RD-full: N = 123) before all final processing and quality control criteria had been applied (see section on EEG Data Collection). Because we include common comorbidities, we performed exploratory/secondary analyses comparing RDs with and without the most common comorbidity, ADHD (accounting for 35.0% of the RD-full sample prior to quality control). It is of interest whether reading deficits manifest differently as a function of attentional deficits, both in regards to behavioral phenotypes and endophenotypes (e.g., EEG profiles). Deficits in attention may affect academic performance across multiple domains, including reading, and some have even proposed that comorbid RD and ADHD constitute a unique 'subtype' of reading disorder that is attributable to attentional deficits [6].

Missing socioeconomic status (SES) data was imputed using the multiple imputation (MI) method as described in [36] in order to preserve statistical power. The MI method comes pre-implemented in SPSS, with various options for model types for scale variables (linear regression or predictive mean matching). The multiple imputation method based on predictive mean matching was chosen to preserve population variance, as previous studies have shown that SES is correlated with multiple measures of cognitive functioning, including reading achievement [37, 38]. SES values coded for self-reported annual income on an ordinal scale of 0 ($< \$10,000$/year) to 11 ($> \$150,000$/year). Prior to imputation, N = 254 (80.6%) participants had SES data available, meaning that a total of 61/315 SES scores were imputed. Missing SES data was imputed using the variables FSIQ, TOWRE PDE scores, and group (RD vs. TR status) as predictors, as these were significantly correlated with the existing SES data.

### EEG data collection

**Resting-state paradigm.** Participants sat in front of a computer and viewed a fixation cross displayed in the center of the screen (Fig 1). They were given the following instructions: 'Fixate on the central cross. Open or close your eyes when you hear the request for it. Press to begin.' During the paradigm they were instructed by a recorded voice from a female research assistant to open or close their eyes at various points throughout the run. A total of 10 segments were collected with eyes open (EO) or closed (EC) at alternating times (5 segments for both the EO and EC conditions). Each EO segment was 20 seconds long; each EC segment was 40 seconds long. Both EO and EC data were used for the analyses (total of 300 seconds). The entire paradigm lasted 5 minutes.

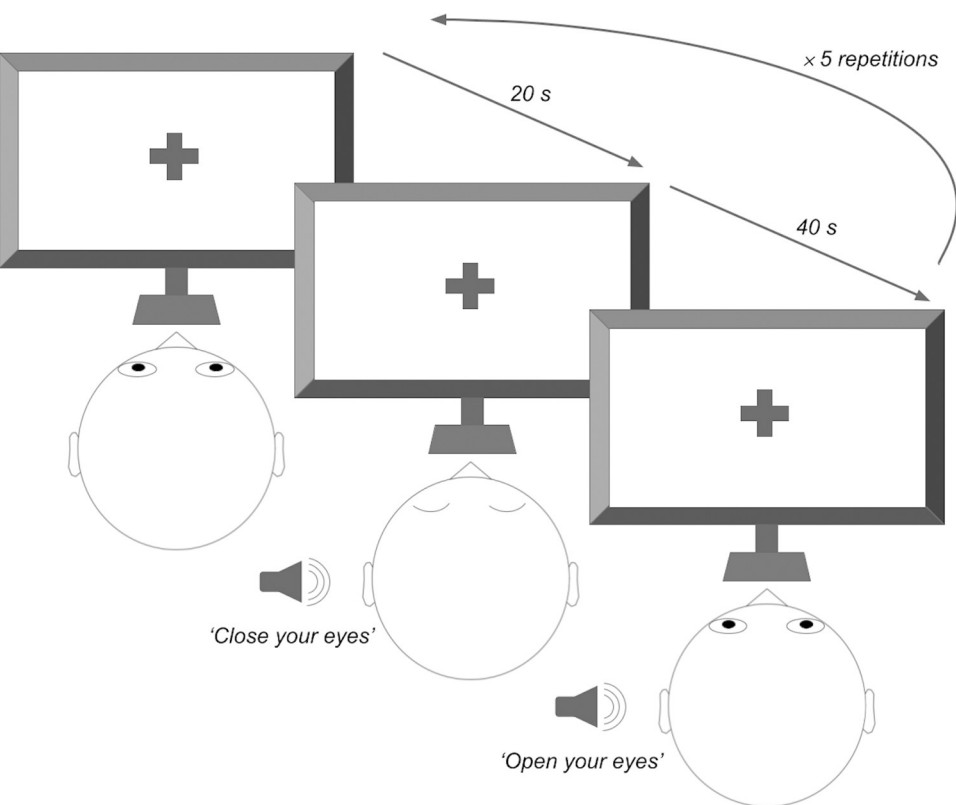

**Fig 1. Illustration of resting-state paradigm.** Participants are asked to close/open their eyes intermittently for a total of 300 seconds.

**Acquisition.** Data were acquired from a high-density 128-channel net using the EGI (Electrical Geodesics, Inc.) Geodesic Hydrocel system. Collection was performed in a sound-attenuated room with a sampling rate of 500 Hz and on-line 0.1–100 Hz bandpass filter. The recording reference was Cz. Electrodes were pre-prepared before placement. Electrode impedance was checked prior to recording and kept below 40 kΩ throughout, being tested every 30 minutes; saline was added if needed [25]. Simultaneous eye tracking was performed by recording eye position and pupil dilation with an infrared video-based eye tracker at a sampling rate of 120 Hz. The 9 EOG channels were placed on the forehead, outer and inner canthi (electrodes E8, E14, E17, E21, E25, E125, E126, E127, and E128).

**Preprocessing and quality control.** All data was preprocessed using Automagic, a MATLAB-based toolbox and EEGLAB plugin for automatic preprocessing and quality assessment of large-scale EEG data [39]. First, individual bad channels were identified and all had line noise removed using the PREP pipeline [40]. A temporary off-line high-pass 2 Hz filter was used. It has been suggested that a high-pass filter of 1–2 Hz produces ideal signal-to-noise ratio when utilizing Independent Component Analysis (ICA), and that this approach often works better than electrooculogram-based ocular artifact regression [41]. ICA was done with ICLabel [42]. Artifacts associated with muscle, heart, and eye activity were removed. An off-line high-pass 1 Hz filter and low-pass 55 Hz filter were also selected. No notch filter was selected.

As part of the quality control process, individual channels flagged by PREP and channels with excessively high or low variance were discarded and later interpolated. PREP flags channels based on the following criteria: high amplitudes with a Z-score >5; a maximum

correlation < .4 with any other channel (within a time window of 1 s); predicted channel activity [based on a random 25% of (so far) 'good' channels] that has a correlation < .75 with the true channel activity, within a certain fraction (>.4) of non-overlapping 5 s time windows; and unusually high-frequency noise, indicated by the ratio between the median absolute deviations of the high-frequency signal (>50 Hz) versus the low-frequency signal having a Z-score >5 [40]. Residual bad channel detection was run after the entire preprocessing pipeline finished to catch any remaining channels with excessively high or low variance. A participant's EEG data was also flagged based on a ratio of high-variance time points exceeding .20, and individual channels were flagged based on overall high amplitude. Criteria for excessively high/low variance were voltage thresholds defined as a function of the standard deviation of the voltage measures across all time points or channels, $25 \times SD(mV)$. The criterion for overall high amplitude was a 40-mV threshold for absolute voltage magnitude. Bad channels were interpolated using the spherical method, which involves projecting all channels to a unit sphere and mapping the identified 'bad' channels to surrounding good channels to estimate missing EEG data [43]. This is one of the most popular methods for bad-channel interpolation, although it may perform less well if there are many bad channels adjacent to one another.

After all preprocessing in Automagic was complete, 338 participants' files were loaded into MNE-Python [44]. Both EO and EC resting-state data were re-epoched into 1-second epochs and had their last second discarded, for a total of up to 300 1-s epochs per subject. Bad epochs were identified and 100 'good' epochs used as the minimum for subject inclusion; only good epochs had their gamma power calculated (see next section). Standards for epoch rejection were a maximum peak-to-peak signal amplitude (PTP) of 100 μV, and a minimum PTP of 1 μV. Of this sample, 23 subjects had at least one missing TOWRE subtest score upon examination for further analysis and were therefore excluded, reducing the sample to N = 315. Finally, the quality control and minimum-epoch criteria were applied, excluding an additional 53 participants (N = 262). A profile of these excluded participants is provided in Supporting Information (S4 Table). Outliers were excluded for each dependent variable in their respective analysis. See Fig 2 for full details of the preprocessing and exclusionary criteria applied to the final sample.

**Calculation of low-γ power.** For this study temporal electrodes were selected a priori due to their relevance to the reading and language network. In addition, some adult resting-state literature reports that intrinsic hemispheric asymmetries in low-γ power may be localized to auditory cortex and/or superior temporal cortices [12, 13]. Electrodes on the outer rim and nasion were excluded. The final sets of electrodes were the following: 'E34', 'E35', 'E39', 'E40', 'E41', 'E45', 'E46', 'E47', 'E50', 'E51', and 'E58', for the left hemisphere; and 'E96', 'E97', 'E98', 'E101', 'E102', 'E103', 'E108', 'E109', 'E110', 'E115', and 'E116' for the right hemisphere (Fig 3).

For each channel, power spectral density (PSD) was estimated using the multitaper method as implemented in the MNE-Python package. The PSD was used to calculate mean gamma band-power in the 30–45 Hz range for all epochs separately; gamma band-power was then averaged across all epochs; and this average value was finally subjected to a log transform of 10 * $\log_{10}(x)$ to derive absolute power in units of $\mu V^2$/Hz (dB). The total number of epochs per participant ranged from 100–300 (median = 225 epochs) and did not differ as a function of RD status (t-test, p = .664) or ADHD status within RDs (p = .903). The average channel values were then additionally averaged across all channels within the left and right hemisphere temporal lobes (separately for each hemisphere). The rationale for choosing the range 30–45 Hz was based on previous findings of RD abnormalities in low-gamma frequencies within this range, and to therefore remove irrelevant and potentially spurious higher-frequency gamma rhythms [12, 18].

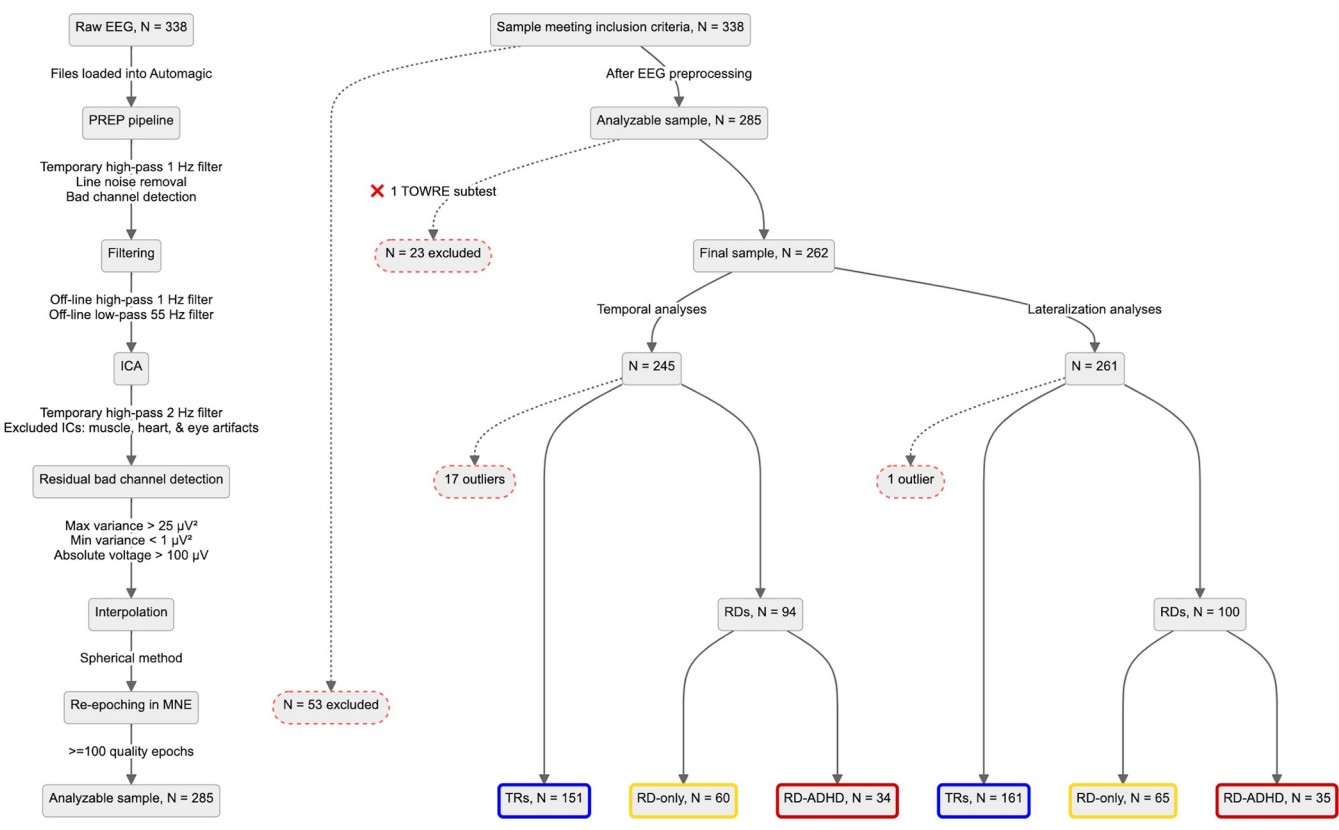

**Fig 2. Flowchart detailing all steps of EEG preprocessing and exclusionary criteria.**

## Analyses

**Outliers.** Outlier analysis and exclusion were performed in SPSS Statistics v28. Outliers were determined to fall outside the range [1st quartile—(1.5 × IQR), 3rd quartile + (1.5 × IQR)], where IQR is the interquartile range, therefore being much higher or lower than the rest of the data. The resulting distributions of each dependent variable were visually examined and judged to be approximately normal. Q-Q plots are presented in Supporting Information (S1 Fig). This left a final N = 245 (17 outliers removed) for the temporal analyses and N = 261 (1 outlier removed) for the temporal lateralization analyses. For both repeated measures (RM) ANCOVAs, Box's Test for equality of covariance matrices and Levene's Test for equality of error variances were run in SPSS prior to the analyses. Only Levene's Test was used for the non-RM ANCOVA and regression models. An alpha threshold of .05 was used for all frequentist significance tests (Box's and Levene's), and effect size was calculated as $\eta^2$.

**Bayesian statistics.** Bayesian analysis, or Bayes factor analysis, is based on Bayes' Rule [45]. Using Bayes' rule, we can state that the probability of a hypothesis H given the observed data is proportional to the probability of observing the data given that H is true, multiplied by the prior probability of H: $P(H \mid \text{data}) \propto P(\text{data} \mid H)P(H)$. Bayesian analysis has become increasingly popular in recent years: one of the basic advantages of Bayesian analysis as opposed to frequentist approaches is that it allows researchers to directly quantify the probability of a given hypothesis, rather than calculating the likelihood of observing the data under the assumption of a particular hypothesis; another is that the relative predictive performance of two competing models can be compared [45].

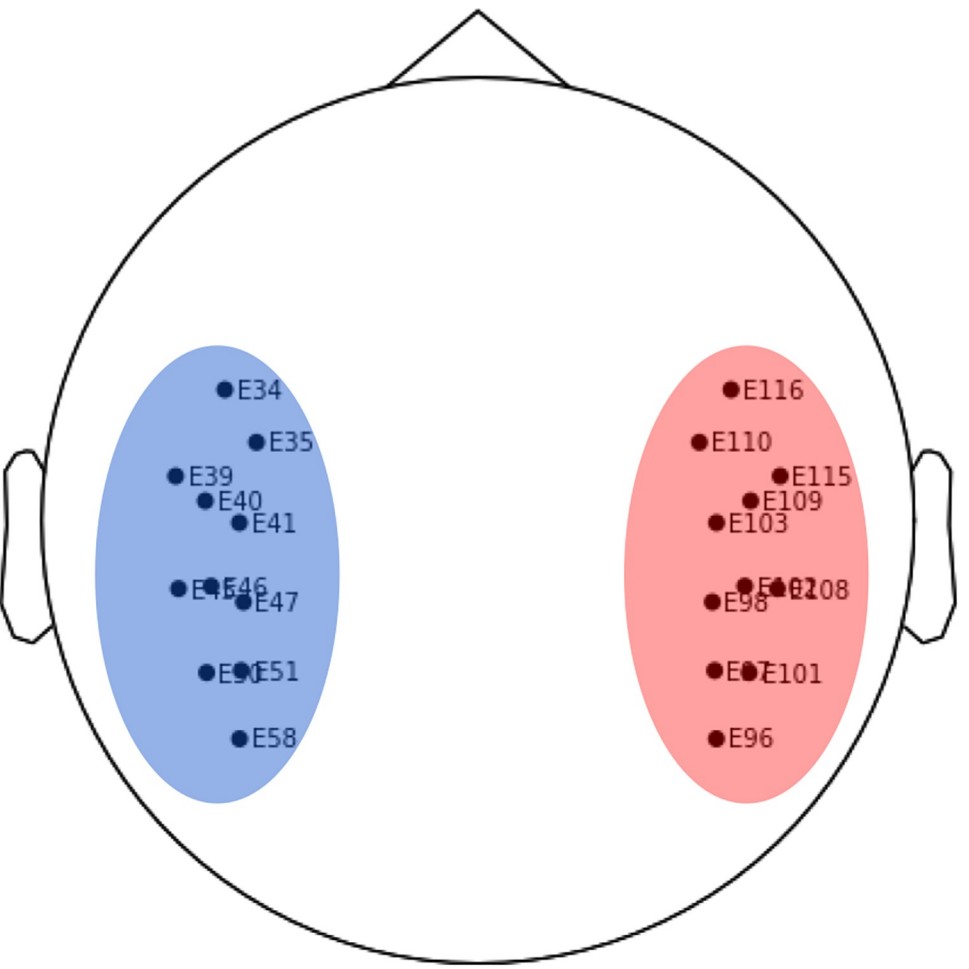

**Fig 3. Chosen electrodes for analyses.**

Bayes factor analysis was conducted in JASP v0.14.1 (jasp-stats.org). The Bayes factor ($BF_{10}$) is the ratio between the probabilities of observing the data given that the alternative hypothesis ($H_1$) is true, and observing the data given that the null hypothesis ($H_0$) is true. A $BF_{10} < .300$ indicates moderate evidence favoring the null hypothesis, while a $BF_{10} > 3.000$ indicates moderate evidence in favor of the alternative hypothesis; values in between indicate that evidence is either anecdotal ($.300 < BF_{10} < 3.000$) or absent, $BF_{10} \approx 1$ or $\log(BF_{10}) \approx 0$ [45]. The inclusion Bayes factor ($BF_{incl}$) is also calculated for all individual factors in each model to determine the contribution of individual effects. The $BF_{incl}$ estimates an effect's unique contribution by comparing all models that contain the effect to equivalent models without that effect.

**Analysis plan.** All preliminary statistical checks (Box's test and Levene's test) and tests of demographics/behavioral data were performed in SPSS v28.0.0.0. All Bayesian analyses were performed in JASP v0.14.1. For analyses where a comparison between the TR and RD-full groups was of interest, post-hoc Bayesian analyses were conducted for each RD subgroup if there was a meaningful main effect of or interaction involving the factor Group, such that the effect was 2x as likely to be included as excluded [BF > 2.000, equivalent to a $\log(BF) > .700$]. Where discussed, Bayes factors are also presented in the form $\log(BF_{10})$ for clarity; positive values indicate greater evidence in favor of the experimental hypothesis, while negative values

indicate evidence in favor of the null. Bayesian estimates of effect size are given as either model averaged posterior distribution values (δ, for ANCOVA/t-test) *or posterior coefficients (β, for regression), with 95% credible intervals (CIs).*

1.1. To examine group differences in temporal low-γ power between TRs and RDs and their interactions with hemisphere, we use a repeated-measures analysis of covariance (RM-ANCOVA) with Hemisphere as a within-subjects factor and Group (TR, RD-full) as a between-subjects factor. Confounding variables were age, SES, performance (nonverbal) IQ/ PIQ, sex, and handedness.

1.2. To examine individual differences in temporal gamma power across decoding skills, associations between TOWRE PDE scores and low-γ power were analyzed with two separate multiple linear regression analyses collapsed across group: TOWRE PDE was the independent variable, while LH and RH temporal low-γ power were dependent. Collinearity diagnostics were run in SPSS prior to the analysis to determine which confounding variables to retain, in order to ensure there was no excessive collinearity with PDE.

2.1. To examine lateralization of intrinsic low-γ power in the temporal lobe, the lateralization index (LI) for low-γ power was quantified according to the same formula used in [18]: (Left—Right) / (Left + Right). One-tailed one-sample t-tests were run on both groups (TR, RD-full) to determine if low-γ power was left-lateralized in the temporal lobes (LI > 0).

2.2. Hemispheric asymmetries were examined for their relationship to phonemic decoding by regressing the LI on PDE collapsed across groups. Confounding variables are the same as in 1.2.

3.1. To confirm that there are no significant differences between RD-only and RD-ADHD groups in temporal low-γ power in either hemisphere, we perform an exploratory RM-AN-COVA with Hemisphere as a within-subjects factor and Group as a between-subjects factor (RD-only and RD-ADHD groups are the two levels for the between-subjects factor), and temporal gamma power as the dependent variable. Confounding variables were age, SES, PIQ, sex, and handedness.

## Results

### Comorbidity rates and demographics

The frequencies of comorbid diagnoses for all groups are given in Table 1. The statistics in Table 1 are derived from a subsample of N = 261, consisting of those who met both quality control criteria and outlier requirements for the temporal lateralization analyses (see Analyses for more details). Chi-squared tests were performed to determine if the rates of diagnosis for the listed comorbidities differed significantly between the groups: TR vs. RD-full, and RD-only vs. RD-ADHD.

For the TR vs. RD-full comparisons, as expected, differences in diagnostic frequency emerged for ADHD, language disorder, and specific learning disorders of reading, where the RD-full group had significantly greater diagnostic rates than TRs (all ps < .05). Unexpectedly, the reverse effect was observed for Generalized Anxiety Disorder, where TRs were diagnosed at greater rates compared to RDs (p < .05). This could be due to the recruitment strategy used by CMI, which emphasizes community-referred recruitment of those with suspected neurodevelopmental and neuropsychiatric conditions as part of their initiative to build a robust database that "captures the broad range of heterogeneity and impairment" [25]. Thus, the RDs could have been recruited largely based on reading or other academic deficits, while TRs likely would have been recruited at higher rates for non-reading-related symptoms, such as anxiety. In relation to its potential effect on the results, see additional descriptive information on the subsamples affected by the more common comorbidities, such as language disorder and

**Table 1. Frequencies of comorbid diagnoses (N = 261).**

| | TR | RD-full | $X^2$ (p) | RD-only | RD-ADHD | $X^2$ (p) |
|---|---|---|---|---|---|---|
| *ADHD Subtypes* | | | | | | |
| ADHD-Combined Type | 0 | 18 | **31.13 (< .001)** | 0 | 18 | **40.77 (< .001)** |
| ADHD-Inattentive Type | 0 | 16 | **27.44 (< .001)** | 0 | 16 | **35.37 (< .001)** |
| ADHD-   Hyperactive/Impulsive Type | 0 | 1 | (.383)[a] | 0 | 1 | (.350)[a] |
| *Specific Learning Disorders* | | | | | | |
| Reading | 0 | 37 | **69.41 (< .001)** | 23 | 14 | .208 (.648) |
| Mathematics | 3 | 3 | (.678)[a] | 2 | 1 | (1.000)[a] |
| Written Expression | 1 | 3 | (.159)[a] | 2 | 1 | (1.000)[a] |
| Language Disorder | 8 | 14 | **6.52 (.011)** | 8 | 6 | (.553)[a] |
| Speech Sound Disorder | 3 | 0 | (.288)[a] | 0 | 0 | |
| *Anxiety Disorders* | | | | | | |
| Generalized Anxiety Disorder | 15 | 2 | **5.42 (.020)** | 2 | 0 | (.540)[a] |
| Specific Phobia | 8 | 4 | (1.000)[a] | 4 | 0 | (.295)[a] |
| Separation Anxiety | 7 | 3 | (.746)[a] | 3 | 0 | (.550)[a] |
| Social Anxiety (Social Phobia) | 8 | 1 | (.160)[a] | 0 | 1 | (.350)[a] |
| Other Specified Anxiety Disorder | 10 | 3 | (.381)[a] | 2 | 1 | (1.000)[a] |
| *Depressive Disorders* | | | | | | |
| Persistent Depressive Disorder (Dysthymia) | 1 | 0 | (1.000)[a] | 0 | 0 | |
| Major Depressive Disorder | 2 | 0 | (.525)[a] | 0 | 0 | |
| Other Specified Depressive Disorder | 1 | 0 | (1.000)[a] | 0 | 0 | |
| *Conduct Disorders* | | | | | | |
| Oppositional Defiant Disorder | 0 | 0 | | 0 | 0 | |
| Intermittent Explosive Disorder | 0 | 0 | | 0 | 0 | |
| Other Specified Disruptive, Impulse-Control, and Conduct Disorder | 0 | 0 | | 0 | 0 | |
| *Other Attentional Issues Not Meeting ADHD Criteria* | | | | | | |
| Unspecified Attention-Deficit/Hyperactivity Disorder | 3 | 0 | (.288)[a] | 0 | 0 | |
| Other Specified Attention-Deficit/Hyperactivity Disorder | 12 | 4 | 1.28 (.258) | 4 | 0 | (.295)[a] |
| Adjustment Disorders | 4 | 1 | (.652)[a] | 0 | 1 | (.350)[a] |

ADHD, attention deficit hyperactivity disorder.

All p-values are two-sided. Empty cells could not have a statistic computed due to both groups having a diagnostic frequency of 0 for that disorder.

[a] Chi-squared tests with expected cell counts <5 report two-sided *p*-values from Fisher's Exact Test.

anxiety disorders, are provided in Supporting Information (S1–S3 Tables). For the RD-only vs. RD-ADHD comparisons, the RD-ADHD group had a self-evidently higher rate of ADHD diagnoses, while all other comparisons were non-significant (ps >.1).

Table 2 describes demographic and behavioral variables, again for the N = 261 sample for the temporal lateralization analyses. The TRs and the RD-full group differed on SES [Mann-Whitney U = 11109.50 (p < .001), Cohen's d = .380, 95% CIs (.250, .496)], with the typical readers coming from families with a higher annual income, consistent with prior literature on the relationship between SES and reading ability [37, 38]. The TRs also performed better (ps < .001) on both FSIQ [Cohen's d = 1.221, 95% CIs (.949, 1.491)] and PIQ [Cohen's d = .534, 95% CIs (.280, .787)], as well as–self-evidently–both TOWRE subtests [Cohen's d = 3.587, 95% CIs (3.189, 3.983) and 3.494, 95% CIs (3.101, 3.883) for PDE and SWE, respectively]. Follow-up comparisons between the RD-only and RD-ADHD subgroups revealed no significant demographic or behavioral differences (all ps > .05). These results support the use of both WISC PIQ and SES as confounding variables for the analyses.

**Table 2. Demographics of sample (N = 261).**

| | TR | RD-full | Test stat. ($p$) | RD-only | RD-ADHD | Test stat. ($p$) |
|---|---|---|---|---|---|---|
| Age, years | 9.2 (1.8) | 9.5 (1.7) | t(259) = -1.40 (.163) | 9.4 (1.7) | 9.7 (1.8) | t(98) = -0.86 (.393) |
| Sex, % female | 48.4 | 44.0 | $X^2$(1) = .49 (.484) | 47.7 | 37.1 | $X^2$(1) = 1.03 (.311) |
| SES | 10.00 | 7.00 | **Mann–Whitney U = 11109.50 (< .001)** | 7.00 | 7.00 | Mann–Whitney U = 1035.50 (.458) |
| Dominant hand (% right) | 88.8 | 88.0 | $X^2$(1) = .04 (.840) | 83.1 | 97.1 | $X^2$(1) = 4.26 (.052)[b] |
| FSIQ | 105.7 (14.6) | 88.7 (12.7) | **t(259) = 9.59 (< .001)** | 89.7 (12.4) | 86.8 (13.3) | t(98) = 1.09 (.278) |
| PIQ, block design | 10.2 (2.9) | 8.7 (2.9) | **t(259) = 4.19 (< .001)** | 8.8 (2.9) | 8.4 (2.9) | t(98) = 0.67 (.508) |
| *TOWRE (scaled scores)* | | | | | | |
| PDE | 107.4 (11.4) | 70.1 (8.4) | **t(251.663) = 30.22 (< .001)[a]** | 69.9 (8.9) | 70.6 (7.6) | t(98) = -0.42 (.678) |
| SWE | 110.6 (12.3) | 71.3 (9.4) | **t(248.129) = 29.19 (< .001)[a]** | 70.5 (9.2) | 72.7 (9.6) | t(98) = -1.14 (.256) |

SES, socioeconomic status (annual income); FSIQ, full-scale IQ; PIQ, performance/nonverbal IQ; TOWRE, Test of word reading efficiency; PDE, phonemic decoding efficiency; SWE, sight-word efficiency.

All p-values are two-sided. Cell values are MEAN (SD), with the exception of SES, which are medians, and sex/handedness, which are percentages.

Hand dominance was determined using the Grooved Pegboard Test.

[a] Equal variances not assumed (failed Levene's Test).

[b] Chi-squared tests with expected cell counts <5 report two-sided p-values from Fisher's Exact Test.

## Temporal low-γ band-power

**RD-full vs. TR analyses.** When analyzing band-power in the RD-full and TR groups, Box's test was not significant (F(3,1615099.27) = 1.526, Box's M = 4.622, p = .206). Both LH and RH temporal gamma power passed Levene's test (LH: F(1,243) = 1.843, p = .176; RH: F(1,243) = .166, p = .684). There was no effect of the covariate age, $\log(BF_{incl\text{-}Age})$ = -.423, when examining data from both groups. Contrary to the hypotheses, when the confounding variables were added to the null model, results from the Bayesian RM-ANCOVA did not reveal a sufficiently large Bayes factor for any of the alternative models with the effects of interest: not Group ($\log(BF_{10})$ = -.889), Hemisphere (-1.567), nor the model with both of the additive effects plus the interaction of Group x Hemisphere (-4.129) (Fig 4). Inclusion Bayes factors were similarly low: $\log(BF_{incl\text{-}Group})$ = -.821 [δ = .048, 95% CIs (-.035, .133)], $\log(BF_{incl\text{-}Hem})$ = -1.564 [δ = .023, 95% CIs (-.015, .060)], and $\log(BF_{incl\text{-}Group\ x\ Hem})$ = -1.693 [δ = .012, 95% CIs (-.026, .049)]. The posterior model probability was also highest for the null model with confounding variables only (excluding age): $P(H_0 \mid data)$ = .344 > $P(H_i \mid data)$ for all alternative models $H_i$. The model with the second-highest posterior probability was the model with all confounding variables including age, $P(H_{0+Age} \mid data)$ = .225.

**Regression of LH/RH low-γ power on phonemic decoding scores.** After running collinearity diagnostics with PDE, included confounding variables were sex (Pearson's $r$ = .066, p = .306) and handedness ($r$ = -.024, p = .710): the planned confounding variables PIQ ($r$ = .319, p < .001, VIF = 1.175), SES ($r$ = .398, p < .001, VIF = 1.226), and age ($r$ = -.126, p = .049, VIF = 1.084) were excluded due to high collinearity with PDE (VIF = variance inflation factor).

LH: The model with the best relative predictive performance included only Sex, $\log(BF_{10})$ = 8.771 (females < males), $\log(BF_{incl\text{-}Sex})$ = 7.984 [β = -.418, 95% CIs (-.589, -.237)]. After observing the data, the odds of including the effect of PDE decreased relative to the prior, $\log(BF_{incl\text{-}PDE})$ = -1.053 [β = -4.725e-4, 95% CIs (-.005, 6.488e-4)]; its marginal inclusion probability also decreased, suggesting there is likely no unique contribution of PDE to left temporal low-γ power. There was a similar lack of effect for handedness, $\log(BF_{incl\text{-}Handedness})$ = -1.315 [β = .007, 95% CIs (-.096, .234)] (Fig 5).

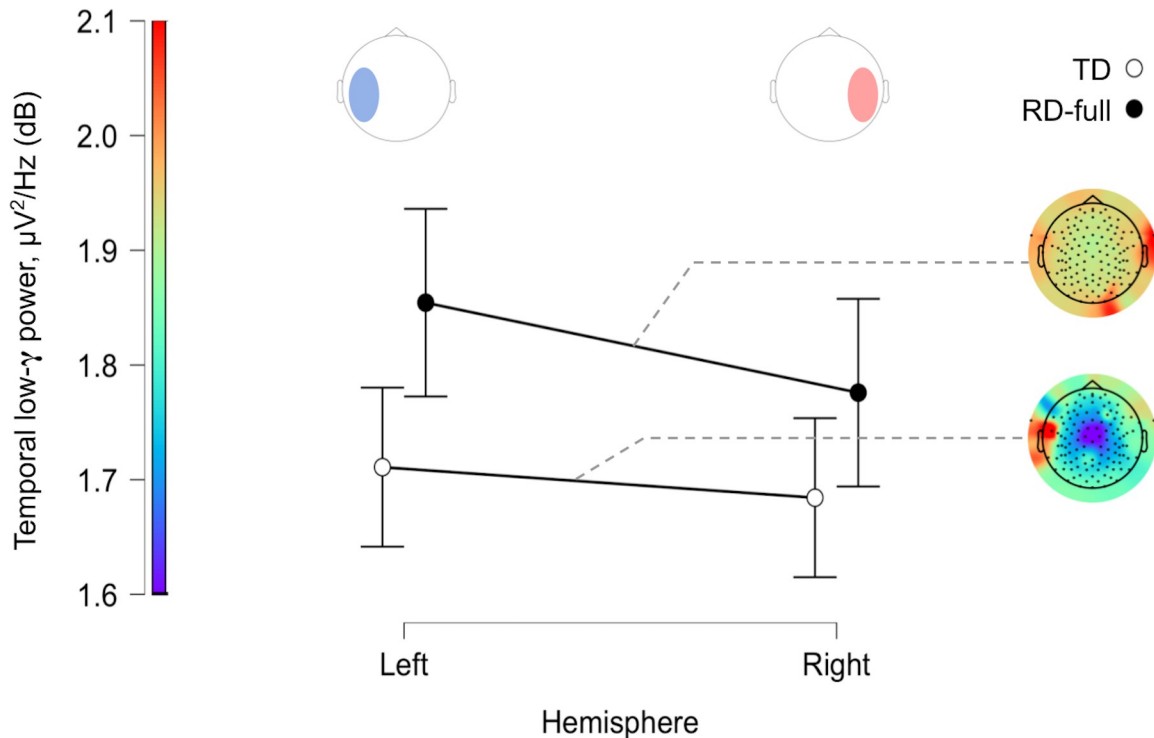

**Fig 4. Plots of temporal low-γ power from both hemispheres in the TR and RD-full groups.** Error bars are 95% confidence intervals. TRs and RDs use the same scale for their group-averaged topographic maps.

### Lateralization index

**Lateralization indices in each subgroup.**   Results indicated slightly different lateralization indices between subgroups. A Bayesian one-sample one-tailed t-test in the positive direction generates values for $BF_{+0}$ and its inverse, $BF_{0+}$, where $BF_{+0} = 1 / BF_{0+}$. Both describe the relative likelihoods of the alternative hypothesis (+, the hypothesis where LI > 0, indicating left-lateralization at rest) and the null (LI = 0, no lateralization at rest).

TRs: A Bayesian one-sample one-tailed t-test resulted in a $\log(BF_{+0}) = -1.155$ for the typical readers. This indicates that the observed data are approximately $1 / e^{-1.155} = e^{1.155} = 3.173x$ as likely under the null hypothesis (no lateralization) for the control group. Estimates of effect size had a median value of $\delta = .104$, 95% CIs (.008, .250).

RD-full: An identical one-sample one-tailed Bayesian t-test generated a $\log(BF_{+0}) = -.064$ for the RD-full group, suggesting equal relative likelihood $[\log(BF_{+0}) \approx 0]$ of the null and alternative hypotheses (anecdotal evidence). Estimates of effect size had a median value of $\delta = .176$, 95% CIs (.022, .367).

RD-only (exploratory): We perform a post-hoc follow-up analysis excluding the comorbid RD-ADHD participants due to the ambiguous result reported for the RD-full group. In contrast to the previous findings, results in the RD-only group generated a $\log(BF_{+0}) = .507$, suggesting that the data are $e^{.507} = 1.660x$ more likely under the alternative hypothesis of left-lateralization than the null. Estimates of effect size had a median value of $\delta = .239$, 95% CIs (.035, .478).

In summary, given the observed data, the null hypothesis of absent resting-state lateralization for low-γ power was moderately better at predicting the observed data for the TRs than the alternative (left-lateralization). Evidence for the RD-full group was not conclusive for

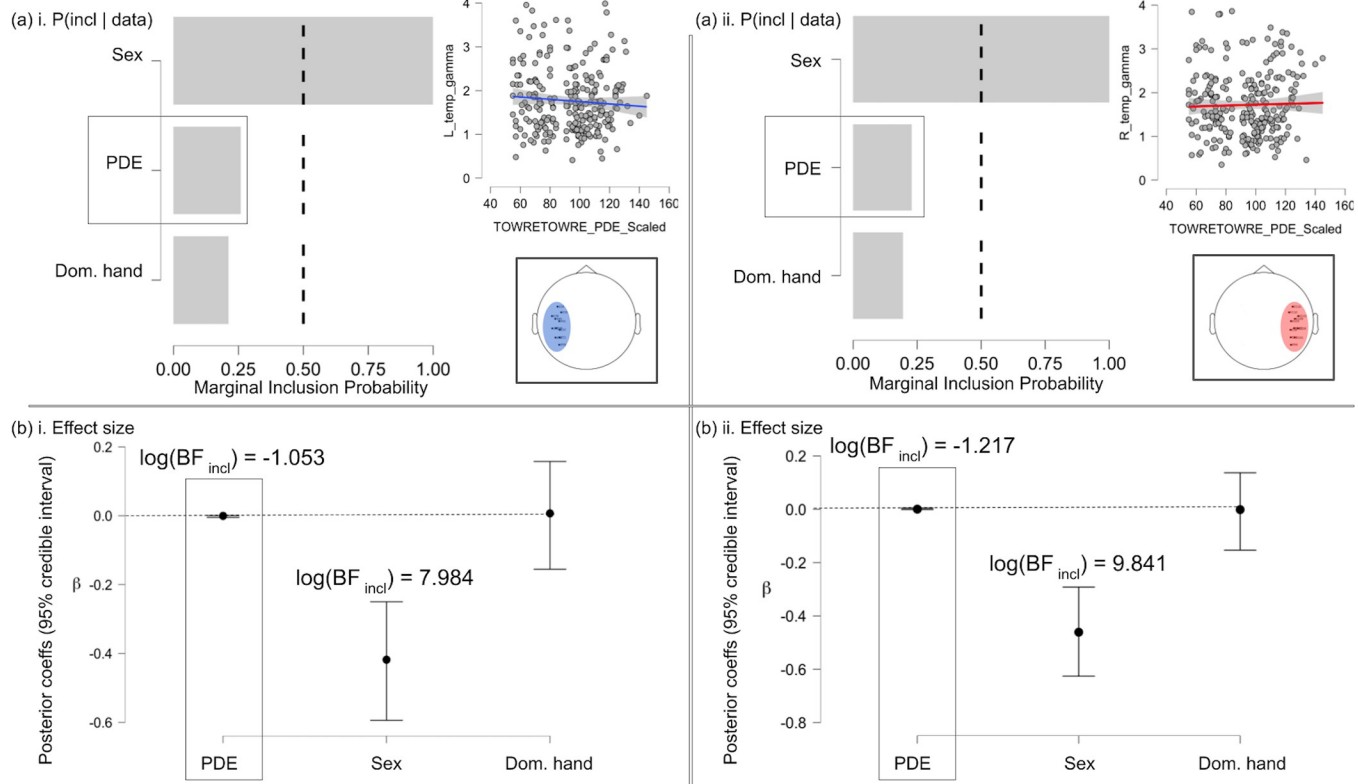

**Fig 5. Descriptive plots, marginal inclusion probabilities [P(incl-E | data)], and posterior coefficients (Bayesian measures of effect size).** (a) Marginal inclusion probabilities (MIPs) for all effects from separate regression models for the left (a-i) and right (a-ii) hemispheres; MIPs reflect the value of the updated priors (initial values of .50, reflecting a 50–50 chance of inclusion before observing data) for each individual effect after observing the data. (b) Measures of effect size (β coefficients) for the left (b-i) and right (b-ii) hemispheres. The independent variable of interest (PDE) is emphasized despite showing no effect. Error bars are 95% CIs. RH: Once again, the most predictive model included Sex (females < males) and no other predictors, $\log(BF_{10})$ = 10.631, $\log(BF_{incl-Sex})$ = 9.841 [β = -.461, 95% CIs (-.626, -.278)]. There was likely no contribution of PDE, $\log(BF_{incl-PDE})$ = -1.217 [β = 3.514e-4, 95% CIs (-7.978e-4, .005)], or handedness, $\log(BF_{incl-Handedness})$ = -1.417 [β = -.001, 95% CIs (-.119, .239)], to right temporal low-γ power.

either the null or alternative hypothesis, $\log(BF) \approx 0$. The subsequent test in the RD-only group suggested, in contrast, that the alternative hypothesis of LI > 0 was comparatively better at explaining the data than the null for this subgroup (Fig 6). Robustness checks based on changes in the specification of priors, and sequential analysis based on the number of observations (accumulated evidence), are provided in Supporting Information (S2 and S3 Figs).

**Regression of hemispheric asymmetry on PDE.** Included confounding variables were sex and handedness. None of the alternative models had a sufficiently large Bayes factor to support their relative likelihood over the null model. The best-performing model relative to the null was the one which included phonemic decoding skill as the sole predictor of LI, $\log(BF_{10})$ = -.876; all others had $BF_{01}$s > 3.000.

*Exploratory post-hoc regression analyses.* The TR and RD-full groups were analyzed separately in an exploratory post-hoc analysis. For the RD-full group, $\log(BF_{incl-PDE})$ was -1.110 and P(incl-PDE | data) < .500, suggesting a lack of a strong relationship between PDE and LI for the RD group. For the TR group, the best-performing model had PDE as the sole predictor with a $\log(BF_{10})$ = .489, meaning that the probability of observing the data was slightly higher under this model than the null (by a factor of $e^{.489}$ = 1.631). For the TRs, but not the RDs, lower PDE scores were associated with left-lateralization, however with a very small effect size estimate of β = -.001, 95% CIs (-.004, .000). (Fig 7). In addition, the $BF_{incl-PDE}$ across all

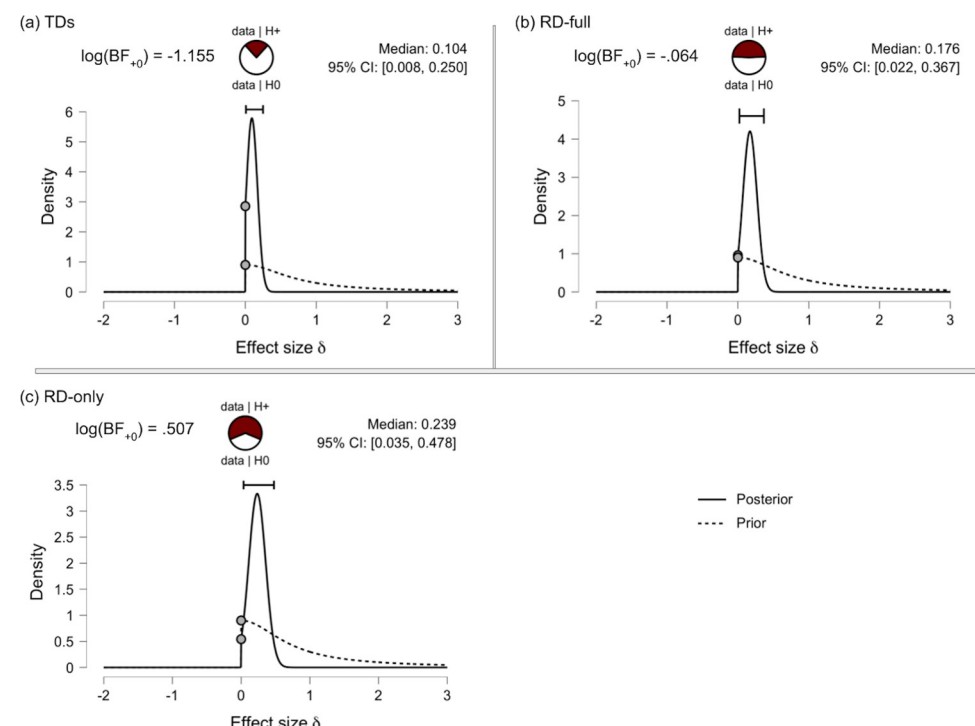

**Fig 6. A visual representation of the Bayesian results from the one-sided one-sample t-tests.** Each plot shows the Bayes factor (upper left) and relative predictive ability of the null vs. alternative hypothesis (upper center). Density plots show both the prior (before updating beliefs) and posterior (after updating beliefs with observed data) of effect size estimates. CIs are 95% credible intervals. Data comes from (a) TRs; (b): RD-full group; (c) RD-only subgroup.

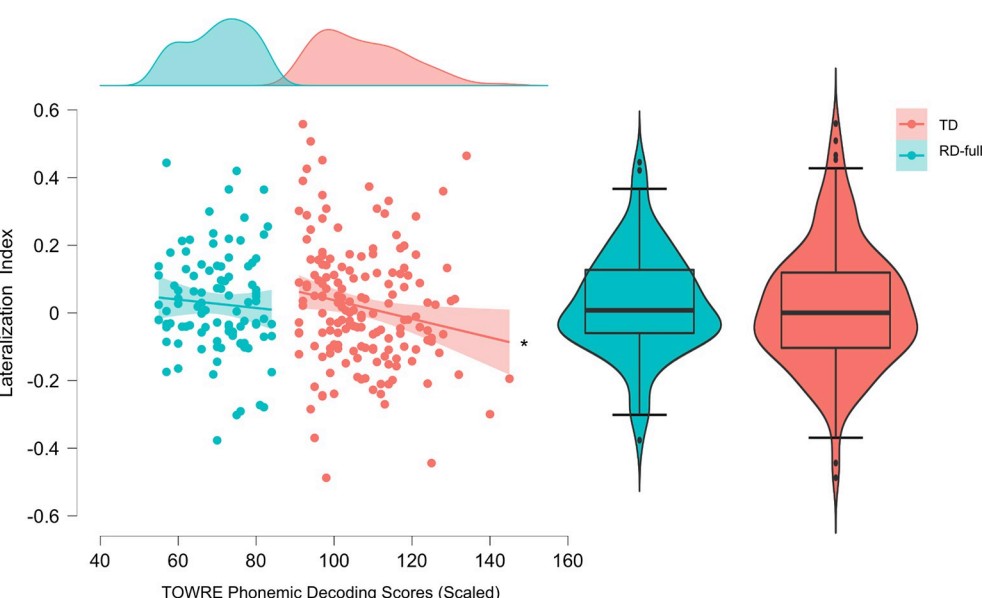

**Fig 7. Descriptive plots illustrating the relationship between TOWRE phonemic decoding and lateralization index (LI).** A star (*) indicates that the $\log(BF_{10})$ for the model including PDE is $> 0$, in favor of the alternative hypothesis. Error bars are 95% confidence intervals.

possible models was still less than 1 (.823), $\log(BF_{incl-PDE})$ = -.194, meaning that P(incl-PDE | data) < P(excl-PDE | data). Given the observed data, the null model was still more likely than the sum of the alternative models that include PDE: $P(H_0 | data)$ = .484 > P(incl-PDE | data) = .452.

### Exploratory: RD-only vs. RD-ADHD analyses

With regards to band-power in the RD-only and RD-ADHD subgroups, Box's test for equality of covariance matrices of the dependent variables was not significant for the RM ANCOVA model (F(3,144979.921) = .362, Box's M = 1.114, p = .781). Both LH and RH temporal gamma power passed Levene's test for equality of error variances across groups (LH: F(1,92) = .008, p = .928; RH: F(1,92) = .484, p = .488).

For the main Bayesian RM-ANCOVA analysis, the confounding variables were age, SES, PIQ, sex, and handedness. There was no effect of the covariate age, $\log(BF_{incl-Age})$ = -.526) on band-power in the temporal lobes for the RD group. When the confounding variables were added to the null model, as expected, all $\log(BF_{10})$ values were sufficiently < 0 for all possible alternative models (Group + intercept, Hemisphere + intercept, Group + Hemisphere + intercept, and Group + Hemisphere + Group*Hemisphere + intercept), such that the data was at least 1.5x more likely to be observed under the null compared to any of the alternative hypotheses. The individual effects of Group, $\log(BF_{incl-Group})$ = -.637 [δ = .061, 95% CIs (-.070, .197)], Hemisphere, $\log(BF_{incl-Hem})$ = -1.089 [δ = .036, 95% CIs (-.022, .093)], and their interaction, $\log(BF_{incl-Group \times Hem})$ = -1.443 [δ = -.008, 95% CIs (-.066, .049)], were unlikely to be included in the model.

## Discussion

The novel contribution of this study was to analyze–in a large, diverse sample–differences in resting-state low-γ band-power associated with decoding-based reading ability and the presence of RD, especially in terms of hemispheric dominance. Additionally, we utilized a Bayesian statistical approach to directly quantify the relative likelihood of our hypotheses compared to the null. Previous research has shown atypical task-based left-lateralization for this frequency range in children with RD, and intrinsic left-lateralization in typical adults. We hoped to address the current gap in research regarding endogenous phonemic-rate neural oscillations in children with RD compared to typical readers, and in particular hemispheric lateralization of low-gamma rhythms and their relationship to functional differences in this population. Contrary to our expectations, in our results we observed both a lack of effect of phonemic decoding ability on low-γ power/lateralization during rest, as well as–counterintuitively–left-lateralization in the low-γ band for the RD-only group, but not the TRs; further discussion of this finding's robustness is warranted.

### RD effects and lateralization

At the group level, there was no evidence of RD status on low-γ power in either the left or right temporal lobes, nor was there an interaction between group and hemisphere. Moreover, although low-γ power in the left hemisphere was larger than in the right across both groups (Fig 4), the effect of Hemisphere did not support a relative likelihood of the alternative hypothesis. The corresponding regression analyses did not show that phonemic decoding scores covaried with low-γ in either the left or right temporal regions. Rather, the Bayes factors generated from our analysis suggest that the observed data is reliably attributable to the null hypothesis. We conclude that at the population level, there are no clear differences in absolute low-γ

power in the temporal regions as a function of phonemic decoding ability. We now discuss this result further in the context of hemispheric lateralization of low-γ rhythms.

Contrary to the hypotheses, the LI did not differ from 0 in the TRs, suggesting an absence of notable leftwards asymmetry in the typically-developing population during rest. The analysis in the RD-full group was less conclusive (BF ≈ 1), with there being no strong evidence for either the null or alternative hypotheses relative to one another. A follow-up test in the RD-only subgroup (excluding ADHD comorbidity) showed relatively stronger support for left-lateralization compared to null effects; this could be interpreted to reflect greater left-dominance of low-γ activity during resting-state in a population with pure phonemic decoding deficits compared to TRs or those with comorbid attentional deficits. However, an additional exploratory post-hoc analysis in the RD-full group suggested that there was no clear covariance between phonemic decoding skills and low-γ lateralization. Therefore, we conclude that any group-level left-lateralization that exists in the RDs as a whole is not strongly linked to individual differences in phonemic decoding skill.

A second exploratory post-hoc analysis suggested that for the TRs, higher phonemic decoding scores were predictive of reduced left-lateralization. However Bayesian statistical tests revealed that despite the observed data being more likely to occur under the alternative hypothesis than the null ($P(\text{data} \mid H_1) > P(\text{data} \mid H_0)$), the null hypothesis was still more likely given the observed data ($P(H_0 \mid \text{data}) > P(H_1 \mid \text{data})$). As a result, we cannot conclude with a high degree of confidence that individual differences in phonemic decoding are associated with low-γ lateralization in our sample during rest. Nonetheless, there are a few interesting points to note from these results.

First, the lack of asymmetry for TRs may indicate that in typical decoders, there is only very minor left-lateralization at rest, if any at all, for hierarchical processing at phonemic timescales in the temporal lobes; as opposed to prior task-based studies suggesting left-hemisphere dominance for various frequency bands [46–49]. Nonetheless, this is consistent with the small effects reported for typical adults in temporal auditory and other cortical language regions at rest [12, 13]. For Morillon et al. [13] in particular, results indicated that speech-sampling frequencies including the low-γ range (<47 Hz) may be expressed asymmetrically primarily during language stimulation and not at rest. We had hypothesized the possibility that in our sample of young children and adolescents, oscillations in the 30–45 Hz range may be more active at rest compared to adults, prior to much of the developmental synaptic pruning and cortical network refinement. At the same time, these children would have undergone some amount of pruning, especially compared to the pre-readers who did have resting-state left-dominance of low-gamma activity [20, 21]. This paper's null findings for lateralization in the controls therefore builds on prior ambiguous resting-state findings from pre-readers and adults. We support the perspective that asymmetric neural oscillations corresponding to phonemic timescales may facilitate linguistic processing primarily during exposure to speech, even at relatively young ages.

Second, the possibility of there being elevated left-dominance in the RD-only group, while contrary to our hypotheses, has been proposed in some previous research in relation to the 'phonemic oversampling' hypothesis of elevated gamma oscillations discussed by Lehongre et al. in [14]. Briefly, the authors hypothesize that both (1) reduced entrainment to acoustic modulations <30 Hz in left auditory cortex, and (2) enhanced entrainment to modulations >40 Hz in both left and right auditory cortex, can account for some of the observed deficits in RD. As our analyzed frequency range included 40–45 Hz (full range 30–45 Hz), our results are not necessarily inconsistent with (2). Lehongre et al. also report reduced left-dominance for entrained—not resting-state—oscillations in the 25–35 Hz range for RD, but do not address the 35–40 Hz range. A similarly unexpected result was reported in [50], whose authors found

that increased cortical synchrony to 20 Hz modulations at ages 5–7 was negatively associated with word/pseudoword reading scores at 2-year follow-ups. There have been multiple other studies which report reduced power in RD within the phonemic frequency range, which is generally defined within the range of 20–50 Hz (see Introduction). We therefore acknowledge that a limitation of both our study and other frequency-based analyses of RD is a lack of consistency (and by extension result generalizability) of the frequency ranges ascribed to cognitive processing categories, particularly phonemic sampling. Regardless, sampling at phonemic rates in non-optimal contexts (for instance, in the absence of speech stimulation, as in this study) may be detrimental to linguistic processing if such 'oversampling' leads to the encoding of environmental noise [51]. Those with RD are reported to be more susceptible to the effects of noise, with many having speech-in-noise perception deficits [52, 53]. Regardless, both the reported statistical results and qualitative examination of the lateralization plots (Fig 7) make clear that intrinsic low-γ power and lateralization are endophenotypes with large individual variability that do not reliably distinguish RDs and TRs at the population level, and we do not want to overstate the degree of effect observed in this study.

Another equally important explanation to consider when evaluating these results critically is participant sampling, in particular sample sizes. An advantage of Bayesian statistical analysis is that it allows one to examine how accumulated evidence (increasing the number of data points sampled) changes the estimated probabilities of the various hypotheses. A figure in Supporting Information demonstrates the effects of such accumulated evidence in our subgroup laterality analyses (S3 Fig). It shows that for equal sample sizes of N = ~50 in both the TR and RD-full subgroups, the accumulated evidence may have implied that the TRs demonstrate anecdotal left-lateralization, while the RD-full group does not. We can therefore appropriately modify our degree of confidence in the reported results regarding group differences by acknowledging this ambiguity.

This paper also contributes to an ongoing discussion on sampling strategies in psychology and neuroscience, especially in relation to the replicability crisis [54]. It is common to have a total N < 50, while at the same time statistical power often goes unacknowledged or unreported. Analyzing large and openly-available datasets may help with this issue somewhat, although it restricts the ability of researchers to analyze functional activity from their own task designs. Combined with the point on generalizability mentioned in the previous paragraph, we also recommend that future studies which aim to replicate or contribute to the neural oscillation literature in RD analyze multiple frequency ranges for each level of processing—for instance, if analyzing phonemic sampling, replicate each analysis for 25–35 Hz, 30–40 Hz, 30–45 Hz, 35–45 Hz, etc. This will allow researchers to investigate whether subtle variation in frequency ranges may produce different results, while removing variance due to differences in data collection and preprocessing procedures.

## ADHD comorbidity

The effects of ADHD comorbidity in our sample are worth some consideration given its high rate of occurrence in the RD population. The extent to which attentional deficits contribute to one of potentially many multifactorial pathways for the onset of reading deficits is still under debate [6]. Our results suggest a potential dissociation between the endophenotypes of 'pure' RD and RD-ADHD, in that left-lateralization of low-γ rhythms for the RDs was only present in the subsample with phonemic decoding deficits and no comorbid attentional deficits. It is possible that in our sample, some proportion of the RD-ADHD participants had secondary rather than primary reading deficits, and may struggle with reading due to 'undertreated' ADHD [6]; for the original description of the 'phenocopy hypothesis', see [55]. In short,

impaired selective attention may negatively impact acquisition of reading skills without a characteristic 'dyslexic endophenotype' of atypical left-hemisphere entrainment; nonetheless, the data presented is insufficient to conclude with high confidence that this is the case for our sample.

Overall, the unique effects of attentional deficits and their relationship to reading ability are an active area of research which generate much controversy. The Multiple Deficit Model of dyslexia suggests that more comorbid conditions can confer liability to more severe deficits in cognitive and behavioral measures [56, 57]. At the same time, it allows one to deduce from the high rates of comorbidity between RD and ADHD that the two disorders may overlap in their etiologies, having some shared explanatory factors that can account for behavioral deficits in both conditions and their comorbidity [58, 59]. Other groups have argued that phase-resetting, entrainment, and the creation of cortical representations for stimuli may be targets of attentional modulation [60–63]. Robust selective attention during early low-level sensory processing could underlie successful entrainment, and this may be affected in children with ADHD [64, 65]. However, the most robust findings of abnormal oscillatory entrainment in ADHD suggest enhanced (rather than attenuated) delta/theta and reduced absolute beta power [65, 66], which we did not examine in this study—therefore, we cannot comment further on these possible effects here. There are also reports of atypical resting-state high-gamma oscillations (61–90 Hz) in adults with ADHD compared to controls [67]. Furthermore, absolute gamma power in the range of 35–45 Hz may be associated with ADHD in children [68], although results are not conclusive (for a review of resting-state EEG studies in neuropsychiatric disorders including ADHD, see [66]).

## Sex effects and their relation to RD/ADHD prevalence

The planned analyses showed that there was a large sex effect on both temporal low-$\gamma$ power and lateralization: female participants had lower gamma power in both temporal gyri and increased leftwards lateralization compared to male participants. These effects held when controlling for phonemic decoding, age, and socioeconomic status. In [69], the authors showed that males 6–13 years old have a greater occurrence of resting-state EEG microstates (defined as short time periods, ~80-120ms, during which the global scalp potential has stable topography) associated with activity in attention networks compared to females in the same age range. A later paper which also examined resting-state EEG microstates, this time in children aged 4–8 years old, further showed that the topographies of sex-dependent microstates (in which males spent more time than females) were largely localized to attention- and cognitive control-related networks [70]. Other researchers' analyses of sex differences revealed that both processing speed and inhibitory control mediated the sex differences often reported in RD, which may also partially explain sex differences in ADHD prevalence [71].

It is possible that the reduced temporal low-$\gamma$ power in females shown here may be reflective of the difference in EEG microstates indicated by past research on sex differences within this age range. However, the full implications of this are unclear. It could be that these differences reflect a developmental difference in attentional processes in boys compared to girls, as described in [71]. We therefore must acknowledge another major limitation of this paper, which is the possibility that, for the Bayesian subgroup t-tests, a higher relative proportion of boys in the RD-ADHD group, and higher relative proportion of girls in the RD-only group, could have affected the result observed. In our sample, the ratio of boys to girls in the RD-only group was close to equivalent (1.1:1); for the RD-ADHD sample, the proportion of boys was higher (1.69:1). Future research should better control for the rates of comorbidity in both males and females while maintaining statistical power, in order to determine whether the

lateralization observed in the RD-only group, but not RD-ADHDs, is attributable in part to sex-based differences. The etiological pathway for these sex differences may overlap with those of attentional networks that are associated with ADHD itself, which could also partially explain the higher rate of ADHD in boys. If this is the case, then maintaining an equal ratio of boys and girls in both the RD-only and RD-ADHD groups could reduce or erase the result observed here. On the other hand, if these sex differences in attention are largely dissociated from the etiological pathway for ADHD one might expect controlling the male:female ratio not to have any effect on the result produced.

## Implications and limitations

The promise of analyzing endogenous rhythms within the brain is that we will one day be able to reliably identify biomarkers within clinical populations without the need for extensive task-based diagnostic batteries. Additionally, such biomarkers will grant us insight into the mechanisms underlying impairment in reading skills and any shared neural substrates with other well-known comorbid conditions. The current study, although its results are counterintuitive and have very small effects, are consistent with the position that there are possible functional differences between those with and without reading deficits during a passive EEG paradigm. However, any such differences must be met with a considerable degree of caution, given the small effects and the lack of replicability within this field. Given the current state of research, we do not advocate for the use of EEG biomarkers in any clinical setting for the identification of RD or dyslexia.

One limitation of our study is that in the analyzed sample we had a considerable reduction in perfectly matched populations in favor of a more 'naturalistic' approach which allows for the occurrence of comorbid conditions (Table 1). While we tried to correct as much for spurious demographic variables as possible and note that rates of most comorbidities did not differ significantly between the TR and RD groups, there is an implicit degree of uncertainty in how the effects of comorbidities might manifest within each subgroup. This is especially true given our findings in the RD-ADHD subgroup, which were distinct from the RD-only subgroup in relation to temporal lateralization. Finally, it is possible that some of the control group participants may have missed receiving what would have been an appropriate RD diagnosis during childhood but had been able to compensate for any deficits (such that they tested within the normal range on TOWRE) by the time they were assessed. This is a concern that exists throughout reading research, particularly with adults, as it is difficult to objectively measure how much one struggled with reading during childhood retroactively. However, given that the current study was done largely with children under the age of 10, we believe that the chance that a high proportion of the controls had successfully compensated for any previous reading difficulties is relatively low.

A second limitation are the challenges involved in analyzing resting-state EEG data. There are a few advantages to using resting-state EEG, chief among which are its relative ease of data collection and cross-study comparability relative to task-based designs. Resting-state EEG also has value for characterizing intrinsic functional activity in the brain. At the same time, there are several challenges involved in analyzing and interpreting this data. The first is that in the realm of spectral analysis of EEG data certain frequency bands–such as alpha–are more commonly studied during resting-state, while the gamma band is relatively less well-studied [66]. The consistency of these results may vary as a function of whether the eyes are kept open or closed: an extensive review of spectral analyses for resting-state EEG in various disorders reported that results from EC studies have greater aggregated consistency than those of EO studies [66]. In this study we attempted to elucidate an 'average' resting-state gamma profile

derived from both EO and EC data, however future studies would do well to examine both states separately with sufficient data for both cases. There is also a degree of 'fuzziness' or over-lap between functional properties ascribed to these different frequency bands: alpha has been reported to be associated with both performance on various intelligence measures and attention [72, 73], while higher-range gamma activity has also been proposed to be related to attention [64, 67], with alpha-gamma coupling being proposed to reflect early perceptual processing [73–75]. These processes are likely to be relevant not just for attention, but for the processing of visual information such as written language.

## Conclusions

The results of our analysis suggest that RD status and poor phonemic decoding do not reliably predict properties of resting-state low-γ rhythms. Results of note only emerged in the post-hoc tests, where RDs with pure decoding deficits and no comorbid attentional disorders showed a lateralization index >0 (left-dominant) for intrinsic low-γ oscillations. Similarly, an explor-atory analysis in TRs revealed that greater phonemic decoding ability was associated with reduced left-laterality. Effect sizes were small, casting some doubt on the robustness of the results.

## Supporting information

**S1 Fig. Q-Q plots of all dependent variables (low-γ power in the left/right temporal lobes and the lateralization index).** Presented data is after exclusion of outliers. Observed values for all variable distributions are compared to an expected normal distribution. More data points which fall along the diagonal indicate greater normality.
(TIF)

**S2 Fig. Robustness checks based on prior specification for the subgroup Bayesian one-sample t-tests.** Plots for each group t-test show how sensitive the Bayes factor is to changes in the initial Cauchy prior width (X-axis). Y-axis indicates the value of the Bayes factor with the given prior; the ordinal scale on the right ('Evidence') is a colloquial interpretation for the cor-responding y-value. Reported results in the main text are derived from the default user prior (gray dot). (a) = TRs; (b) = RD-full; (c) = RD-only subgroup.
(TIF)

**S3 Fig. Sequential analysis (accumulated evidence) of the subgroup Bayesian one-sample t-tests.** These tests indicate how the quantified degree of evidence changes as additional 'sam-ples' are added. X-axis ($n$) indicates sample size. Y-axis reflects the value of Bayes factors at the given sample size; the ordinal scale on the right ('Evidence') is a colloquial interpretation for the corresponding y-value. Line style indicates the value of the Cauchy prior. (a) = TRs; (b) = RD-full; (c) = RD-only subgroup.
(TIF)

**S1 Table. Descriptive information on participant subsamples with specific comorbidities (collapsed across reading ability).** SES, socioeconomic status (annual income). All p-values are two-sided. Age is given in years, with MEAN (SD) format. SES is given as median. Sex and dominant hand are percentages. Frequentist tests significant at the .05 α-level and Bayesian t-tests with a $BF_{10} > 2.00$ [equivalent to a $\log(BF_{10}) > .70$] are bolded. Hand dominance was determined using the Grooved Pegboard Test. The only significant contrast for demographic variables occurred for sex, such that the participants with anxiety had a higher proportion of girls than the participants without anxiety. All comparisons for low-gamma and the

lateralization index favored the null hypothesis with a relative likelihood >2.00 [log(BF$_{10}$) < -.70].
(DOCX)

**S2 Table. Descriptive information on TRs with specific comorbidities.** SES, socioeconomic status (annual income). All p-values are two-sided. Age is given in years, with MEAN (SD) format. SES is given as median. Sex and dominant hand are percentages. Frequentist tests significant at the .05 α-level and Bayesian t-tests with a BF$_{10}$ > 2.00 [equivalent to a log(BF$_{10}$) > .70] are bolded. Hand dominance was determined using the Grooved Pegboard Test. In typical readers, the only significant contrast for demographic variables occurred for sex, such that the participants with anxiety had a higher proportion of girls than the participants without anxiety. All comparisons for low-gamma and the lateralization index favored the null hypothesis with a relative likelihood >1.78 [log(BF$_{10}$) < -.58].
(DOCX)

**S3 Table. Descriptive information on RDs with specific comorbidities.** SES, socioeconomic status (annual income). All p-values are two-sided. Age is given in years, with MEAN (SD) format. SES is given as median. Sex and dominant hand are percentages. Frequentist tests significant at the .05 α-level and Bayesian t-tests with a BF$_{10}$ > 2.00 [equivalent to a log(BF$_{10}$) > .70] are bolded. Hand dominance was determined using the Grooved Pegboard Test. In impaired readers, there were no significant contrasts between demographic variables based on anxiety or language disorder Dx. Similarly, all comparisons for low-gamma and the lateralization index favored the null hypothesis with a relative likelihood >2.00 [log(BF$_{10}$) < -.70].
(DOCX)

**S4 Table. Demographics of included vs. excluded samples based on EEG minimum-epoch criteria.** SES, socioeconomic status (annual income); PIQ, performance/nonverbal IQ; TOWRE, Test of word reading efficiency; PDE, phonemic decoding efficiency; SWE, sight-word efficiency. All p-values are two-sided. Cell values for age and PIQ are MEAN (SD). SES is given as median. RD, ADHD, sex, and handedness are percentages. Hand dominance was determined using the Grooved Pegboard Test. All group comparisons of demographic and behavioral variables between included vs. excluded (based on the minimum 100 'good' epoch criterion) participants, including rates of RD/ADHD diagnoses, are n.s. at the .05 α-level.
(DOCX)

## Author Contributions

**Conceptualization:** Oliver H. M. Lasnick, Fumiko Hoeft.

**Data curation:** Oliver H. M. Lasnick.

**Formal analysis:** Oliver H. M. Lasnick.

**Investigation:** Oliver H. M. Lasnick.

**Methodology:** Oliver H. M. Lasnick, Roeland Hancock, Fumiko Hoeft.

**Project administration:** Oliver H. M. Lasnick.

**Supervision:** Roeland Hancock, Fumiko Hoeft.

**Visualization:** Oliver H. M. Lasnick.

**Writing – original draft:** Oliver H. M. Lasnick.

**Writing – review & editing:** Oliver H. M. Lasnick, Roeland Hancock, Fumiko Hoeft.

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
