## [Decision Letter · Decision Letter 0]

13 Oct 2023

PONE-D-23-28275Bayesian analysis of resting-state temporal low-γ power in children with phonemic decoding deficits and certain comorbidities

PLOS ONE

Dear Dr. Lasnick,

Thank you for submitting your manuscript to PLOS ONE. After careful consideration, we feel that it has merit but does not fully meet PLOS ONE’s publication criteria as it currently stands. Therefore, we invite you to submit a revised version of the manuscript that addresses the points raised during the review process.

I would like to thank your valuable submission. 

This study is indeed commendable, is brimming with potential, and I genuinely enjoyed reading it.

I do have some concerns, and they are a bit lengthy, but they reflect the thoroughness of the study. Although they might sound annoying, they are intended to be helpful somehow.

General

1) Please check grammar and ensure consistency are maintained throughout the paragraphs, along with smoother transitions;

2) Double-check your refs. to ensure they are in the correct Vancouver format;

3) Please check section topics to ensure they are aligned (e.g., Results should have 1. and so on);

4) Maintain consistent decimal styles throughout the text;

Make sure to report effect sizes and CIs;

5) Use brackets before parentheses, not double parentheses;

Title

The Title is adequate and clearly outlines the objectives and provides a descriptive overview. However, I'd recommend that the same suggestions made for the Abstract section be considered in Title.

Overall, while the title is informative, there are a few points to ponder:

1) Is it absolutely necessary to emphasise 'Bayesian' in the title? Perhaps this could be made clear at the very beginning of the Introduction;

2) Using 'γ' in the title may not be immediately clear to all readers. Although I won't insist on changes, it's worth considering the broader audience;

3) Instead of "and certain comorbidities," consider a more specific or easily understandable term for readers;

4) While it's up to the authors', the intriguing findings in your study could be a very good point to consider a more 'punchy' Title.

Consider how others might think: Liking the Title, moving on to the Abstract, and then feeling impulsed to read the entire study;

Abstract:

I believe the abstract is suitable for readers and researchers within the topic. However, if you want to gather a broader audience, you might need to rethink the language. As of now, if you wish to maintain it as is, I have no disagreements. Nevertheless, bear the possibility of reaching broader wider readership due to your valuable study. Here are some aspects to consider:

1) Think about shortening a few parts, your abstract is almost pushing the limits;

2) I'd highly recommend expanding the abstract with a summary and implications of your study, including statements like "Our study... contrary to our expectations." Emphasise that the unexpected findings challenge existing theories and indicate that language processing is more complex than previously thought. From now I can only think on phonemic (i.e. symbols and phonemes), the explanation of Bayesian (i.e. that won't be considered by some) and temporal lateralisation, that some only will gather this after your results.

This after shorten sections;

3) What about begin with a statement that challenges the solidity of past findings to grab even more attention?;

4) Explain the methods for investigating low-γ power;

5) Discuss the implications of the findings for our understanding of dyslexia;

6) Use stats values whenever possible; 

7) Explain the relevance of oscillations/power in the left temporal area.

8) Think about practical applications or potential directions suggested by your findings;

Introduction

1) Consider shortening Intro, it extends for over six para., with para. 5-7 being notably extensive;

2) Avoid introducing more than one idea within a single paragraph. Refine the content while maintaining the flow;

3) Again, considering a broader audience, I'd highly suggest that potentially complex terms be briefly explained within the text, in keeping with the authors' writing style. This will enhance readability;

4) Provide a well-structured presentation of the background, explaining the significance of your study and objetives; 

Methods

This section is very well-written and I'm quite happy with its rigour, clarity, and presentation. I have a few comments to consider:

1) This section needs primarily focus on the core aspects of the study. Information about Tables, sample characteristics, and other details may be more appropriately placed in the Results section;

--- Participants

2) The authors can consider clarifying who is allowed to use the CMI-HBN, offering more details on this aspect;

3) Include the IRB number;

4) Please provide a clear explanation for the chosen age range (13-), as it may not be immediately evident to all readers;

Consider referencing Cellier's study from 2021, which discusses age and wave oscillations, although the age range differs slightly;

5) Include Cronbach values for the WISC and TOWRE. They are known, but providing thiese values ensure alignment to best practices;

6) Explain why z-scores were not utilised;

7) I've noticed something that can be misinterpreted or, if correctly interepreted, a drawback. I'd like the authors to clarify:

The criteria for the TR group, specifically using TOWRE scores >90, may not be the most suitable control for the RD group with TOWRE scores <85. The TR group is identified based on having TOWRE scores significantly higher than the RD group, but this may not adequately control for the presence of comorbidities or other factors that could influence the analysis of both groups. 

The concern is not about using a non-restricted sample but rather the potential lack of a well-matched control. It's important to consider matching demographics or other variables that could make the TR and RD groups more similar for a more robust interpretation and explanation of the findings;

8) Please clarify the criteria and procedures used for achieving a consensus on diagnoses;

9) Explain how differences unrelated to reading may occur due to the lack of a matched TR group;

10) Consider categorising confounding variables as moderators rather than covariates; 

11) Please provide details on the tools and criteria used for diagnosing ADHD and other conditions, along with their Cronbach values;

12) Move the section explaining analyses to its proper location;

13) Place tables and statistics (e.g. line 185) in their respective sections;

14) Please clarity why MI is preferable to maximum likelihood estimation in this context, potentially within the analyses section;

15) Please include additional demographic details, such as gender proportions, IQ values and proportions (i.e. I've only seen IQ), and whether participants' family members or parents had a history of similar conditions;

16) Consider adding a flowchart illustrating, because it's *really/ important, explaining the progression from the beginning, including sample selection, exclusions, and criteria;

17) Clarify how the sample size was reduced from 315 to the final observed number;

18) Please address the issue of clinical diagnoses and their potential influence on the generalisability of findings;

19) Please specify the sampling method;

20) Please consider to xplain all variables measured and observed in the study;

--- EGG

1) Please describe how instructions and the conditions were made uniform for all participants;

2) Clarify the approach to managing challenges in resting state EEG, especially with regard to the presence of alpha waves. As the authors are aware, resting state is where the alpha likes to play. I'd like authors to clarify this, especially when avoiding that some waves bury in others;

3) Considering this, did the authors check the temporal pattern of the waves? Or SNS to avoid buried waves and also to check and minimise individual's differences, which is common in resting state;

4) Discuss the use of ICA and cross-modal aspects, or clarify when these will be reported;

5) Explain how montages were applied and whether electrodes were pre-prepared for placement;

6) Detail the rationale for using the PREP - as the use of Python for preprocessing is being evident. Actually, this is just a matter of curiosity;

7) I am relieved because the authors used a higher-density, so it's a little bit 'easier' to cut artifacts. The montage was set before or just a little bit beforing resting state?;

8) Please clarify the use of a 53 Hz (i.e. notch filter?) and debate epochs (i.e. or why it was divided into two sections);

9) A fluxogram - a sequential fluxogram - would be fantastic here. This explanation for EEG is commendable;

10) Avoid beginning sentences with abbreviations (e.g. line 227);

11) Clarify whether spectral and ERP analyses were conducted;

12) Explain the criteria for identifying individual artifacts, particularly without EMG. The authors checked it manually too?;

13) Channels flagging need to have the limits presented;

14) Please clarify the spherical method and its potential influence on data;

15) Define the criteria for 1-second epochs;

16) ust now the authors mention that partiipanst were removed - but why? Maybe the data would be inflated or diferent with them?;

17) The exclusion of values that exceed certain expected or the mean, is indeed an interesting step in the data preprocessing. However, researchers might want to know more about the specific criteria used for this exclusion. Since your study has a substantial sample, some might argue that the exclusion of these values may not significantly influence the overall finding. Providing further details or explanation for these exclusions could help address potential questions or concerns regarding this aspect of your data processing;

--- Calculation for low-gamma

1) Clarify whether you calculated PSD for each epoch or used a different approach. For example, I got the a'll epochs', but defining a range would be also useful for a better control;

2) Explain the rationale for choosing 45 Hz. This is a matter of curiosity. If you have the opportunity to check Haenschel's study (2000) there's explanation of this question and the one about buried waves;

3) Confirm if MNE automatically used FFT as the algorithm for calculations;

4) Please clarify whether any manual data checking was conducted at each step of the analysis;

--- Bayes

1) If the authors find that the explanation of BF₀ and BF₁₀ is a central focus of their study, they can mainting it. Otherwise, provide a concise explanation or refer readers to relevant sources for more in-depth information. This approach will help maintain the focus on the main research objectives and findings while still providing some context for the Bayesian analyses;

2) Consider using an exponential scale for Bayes factors (BF) to enhance the rigour;

3) Please explain if you explored different models to assess their usability and to support the choice made;

--- Analyses

1) Please clearly state the software used for the analyses. Include effect sizes and CIS, and maintain consistency in the number of decimal places. This will ensure the same rigour;

2) Consider you have a complex data and, consequently, analyses. Using Bayesian for various types of models can indeed help mitigate issues related to multiple analyses and provide a more unified approach. Bayesian offer advantages in handling uncertainty, which can be especially valuable when dealing with complex, multiple data and multiple DVs;

In Bayesian regression, oauthors can observe posterior distribution for model parameters and make probabilistic statements about the relationships between variables. Additionally, Bayesian SEM allows for more flexible modeling of complex relationships among variables, making it suitable for your data. And, important as well, Bayesian MANOVA provides a comprehensive way to analyse data when there are more than one DVs, making it a suitable alternative to  multilevel analyses;

Results

Please use scatter and heatmaps whenever possible, aligning with the rigour of your study.

We look forward to receiving your revised manuscript.

Kind regards,

Thiago P. Fernandes, PhD

Academic Editor

PLOS ONE

Journal Requirements:

4.Thank you for stating the following financial disclosure: "Author OHML was supported by the following NIH grant(s): T32DC017703 and F31HD107944; and the NSF grant NRT-UtB1735225.

Author RH was supported by the following NIH grant(s): R01HD094834.

Author FH was supported by the following NIH grant(s): R01HD094834, R01HD096261, and U24AT011281."

Reviewers' comments:

Reviewer's Responses to Questions

**Comments to the Author**

1. Is the manuscript technically sound, and do the data support the conclusions?

Reviewer #1: Yes

Reviewer #2: Yes

2. Has the statistical analysis been performed appropriately and rigorously? 

Reviewer #1: Yes

Reviewer #2: Yes

3. Have the authors made all data underlying the findings in their manuscript fully available?

Reviewer #1: Yes

Reviewer #2: Yes

4. Is the manuscript presented in an intelligible fashion and written in standard English?

Reviewer #1: Yes

Reviewer #2: Yes

5. Review Comments to the Author

Reviewer #1: The findings are good and writing style is also appreciated. However, too much of variety in the sample types with unequal sample sizes could have potentially led to inconclusive results. Focussing on few common populations where reading disability is encountered may have provided better insights into their neural processing of reading.

Reviewer #2: 1. Introduction (page 3, lines 43-44): It would be beneficial to clarify what exactly “general population” indicates. Is it referring to the global population or to a specific country or region?

2. Introduction (page 3, line 46): It would be better to provide a few examples of neurodevelopmental disorders here.

3. Introduction (page 3, lines 48-49): To accommodate potential readers from different areas, it is suggested to provide brief explanations or definitions for certain terminologies, such as “rapid automatized naming.”

4. Introduction (page 5-6, lines 107-111): Please provide a comprehensive and in-depth discussion of the potential clinical or practical implications that can be derived from this study.

5. Materials and Methods – Participants (page 7, lines 138-139): Please add a reference supporting this statement.

6. Materials and Methods – Participants (page 7, lines 145-146): Please add the considerations for why a cutoff of 70 was chosen for the IQ scale, as opposed to other values.

7. Materials and Methods – Participants (page 8, lines 156-157): Is it possible that not receiving a prior clinical diagnosis of RD was simply because the individual did not go to a doctor, which doesn't mean the subject doesn't have RD?

8. Materials and Methods – Participants (page 8, lines 157-159): As the inclusion of comorbidities could introduce serious biases into the results, one of the major concerns of this study was whether any efforts were made to prevent or decrease their potential influences.

9. Materials and Methods – Participants (page 8, lines 163-166): Although comorbidities were allowed based on consensus diagnoses from clinicians, it is still unclear what considerations led to this decision (i.e., whether a comorbidity was allowed or not).

10. Materials and Methods – Participants (page 8, lines 167 and 170): It would be beneficial to include a profile of background characteristics of the 54 excluded subjects (i.e., 315 – 261) and to assess whether they differ from the included sample in some aspect.

11. Materials and Methods – Participants (page 8, lines 174-175): It is encouraged to conduct supplementary analyses comparing individuals with RDs, both with and without second most common comorbidity as well. Even if limited to descriptive statistics, this would provide a more comprehensive picture.

12. Materials and Methods – Participants (page 8, lines 174-175): Please strengthen the implications regarding the comparison between RDs with and without ADHD in terms of clinical and practical aspects.

13. Materials and Methods – Participants (page 10, lines 185-187): Please discuss this unexpected reverse effect and its potential effects on the results of this study.

14. Materials and Methods – Participants (page 10, lines 189-190): Please explain the approach for testing non-random missingness of data.

15. Materials and Methods – Quality control (page 13, lines 238-239): It could still be unclear for readers how to define excessively high or low variance for channels. It would be beneficial to provide an instance.

16. Materials and Methods – Analyses (page 14, lines 269-270): I would encourage to try to explain more how to visually examine exactly for normal distributions.

17. Materials and Methods – Bayesian statistics (page 14, lines 277-278): Please add a reference for this method adopted.

18. Materials and Methods – Bayesian statistics (page 15, lines 283-285): Since Bayesian approach is the key method of this study, it would strengthen this study’s uniqueness if having a detail comparison between Bayesian approach and frequentist approaches in terms of advantages and disadvantages in advance.

19. Materials and Methods – Analysis Plan (page 16, lines 311-313): Please provide a brief result with statistics on collinearity diagnostics.

20. Materials and Methods – Analysis Plan: It is encouraged to do some sensitivity analyses to strengthen the robustness of this study’s results.

21. Discussion (page 21, paragraph 1): It would be beneficial to include brief summarized statements that reinforce the novel contributions of this study to address the existing knowledge gap.

22. Discussion (where appropriate): It may not be clear what future clinical and practical implications can be derived from this study. Therefore, it would be beneficial to include a paragraph that discusses potential implications.

6. PLOS authors have the option to publish the peer review history of their article (what does this mean?). If published, this will include your full peer review and any attached files.

Reviewer #1: No

Reviewer #2: No

---

## [Author Response · Author response to Decision Letter 0]

24 Nov 2023

First, we would like to sincerely thank both the editor and the reviewers for their thorough and insightful comments. We appreciate your valuable feedback and believe that all of the changes made significantly improved our manuscript. We have made various changes throughout the text, and all changes are visible in the “Revised Manuscript with Track Changes.docx” document. A clean version has also been submitted. 

All major changes/additions associated with a specific reviewer’s comment have been provided in this letter with page/line numbers also provided. These page/line numbers are in reference to the “Revised Manuscript with Track Changes.docx” document, which may need to be opened in Microsoft Word to be viewed in context. However we hope that providing these changes in this letter as well will facilitate the review process.

Editor

General

1) Please check grammar and ensure consistency are maintained throughout the paragraphs, along with smoother transitions;

2) Double-check your refs. to ensure they are in the correct Vancouver format;

3) Please check section topics to ensure they are aligned (e.g., Results should have 1. and so on);

4) Maintain consistent decimal styles throughout the text;

Make sure to report effect sizes and CIs;

5) Use brackets before parentheses, not double parentheses;

We have performed the requested checks as described, and hope that our manuscript is in full compliance with all grammar/style guidelines. We have (1) made various grammatical changes/corrections throughout, (2) reformatted all references in Vancouver format, (3) standardized all section headings, (4) formatted all decimals and added additional effect sizes and CIs, and (5) reformatted brackets/parentheses where applicable.

Title

The Title is adequate and clearly outlines the objectives and provides a descriptive overview. However, I'd recommend that the same suggestions made for the Abstract section be considered in Title.

Overall, while the title is informative, there are a few points to ponder:

1) Is it absolutely necessary to emphasise 'Bayesian' in the title? Perhaps this could be made clear at the very beginning of the Introduction;

2) Using 'γ' in the title may not be immediately clear to all readers. Although I won't insist on changes, it's worth considering the broader audience;

3) Instead of "and certain comorbidities," consider a more specific or easily understandable term for readers;

4) While it's up to the authors', the intriguing findings in your study could be a very good point to consider a more 'punchy' Title.

Consider how others might think: Liking the Title, moving on to the Abstract, and then feeling impulsed to read the entire study;

These are valuable suggestions. It is indeed important that the title reflects an easily-understandable summary of the results of our study. Upon considering the points raised, we suggest the following revised title: “Left-dominance for resting-state temporal low-gamma power in children with impaired word-decoding and without comorbid ADHD”.

Abstract:

I believe the abstract is suitable for readers and researchers within the topic. However, if you want to gather a broader audience, you might need to rethink the language. As of now, if you wish to maintain it as is, I have no disagreements. Nevertheless, bear the possibility of reaching broader wider readership due to your valuable study. Here are some aspects to consider:

1) Think about shortening a few parts, your abstract is almost pushing the limits;

2) I'd highly recommend expanding the abstract with a summary and implications of your study, including statements like "Our study... contrary to our expectations." Emphasise that the unexpected findings challenge existing theories and indicate that language processing is more complex than previously thought. From now I can only think on phonemic (i.e. symbols and phonemes), the explanation of Bayesian (i.e. that won't be considered by some) and temporal lateralisation, that some only will gather this after your results.

This after shorten sections;

3) What about begin with a statement that challenges the solidity of past findings to grab even more attention?;

4) Explain the methods for investigating low-γ power;

5) Discuss the implications of the findings for our understanding of dyslexia;

6) Use stats values whenever possible;

7) Explain the relevance of oscillations/power in the left temporal area.

8) Think about practical applications or potential directions suggested by your findings;

Thank you for the thorough review of our abstract. We agree that shortening the abstract and focusing on the key results will be more compelling to potential readers. We have removed some unnecessary sentences/clauses so that the updated abstract devotes more relative space to the discussion of results. The abstract now reads as follows (264 words):

“One theory of the origins of reading disorders (i.e., dyslexia) is a language network which cannot effectively ‘entrain’ to speech, with cascading effects on the development of phonological skills. Low-gamma (low-γ, 30-45 Hz) neural activity, particularly in the left hemisphere, is thought to correspond to tracking at phonemic rates in speech. The main goals of the current study were to investigate temporal low-γ band-power during rest in a sample of children and adolescents with and without reading disorder (RD). Using a Bayesian statistical approach to analyze the power spectral density of EEG data, we examined whether (1) resting-state temporal low-γ power was attenuated in the left temporal region in RD; (2) low-γ power covaried with individual reading performance; (3) low-γ temporal lateralization was atypical in RD. Contrary to our expectations, results did not support the hypothesized effects of RD status and poor decoding ability on left hemisphere low-γ power or lateralization: post-hoc tests revealed that the lack of atypicality in the RD group was not due to the inclusion of those with comorbid attentional deficits. However, post-hoc tests also revealed a specific left-dominance for low-γ rhythms in children with reading deficits only, when participants with comorbid attentional deficits were excluded. We also observed an inverse relationship between decoding and left-lateralization in the controls, such that those with better decoding skills were less likely to show left-lateralization. We discuss these unexpected findings in the context of prior theoretical frameworks on temporal sampling. These results may reflect the importance of real-time language processing to evoke gamma rhythms in the phonemic range during childhood and adolescence.”

Introduction

1) Consider shortening Intro, it extends for over six para., with para. 5-7 being notably extensive;

We have trimmed some of the excess text from our Introduction while adding additional information where requested by the Reviewers. While it may not be significantly shorter, we have tried to ensure that all background information presented is valuable for helping to understand and justify our study.

2) Avoid introducing more than one idea within a single paragraph. Refine the content while maintaining the flow;

We have reorganized the section slightly, moving some sentences/clauses to more appropriate locations in the interest of keeping each paragraph focused on a single key point.

3) Again, considering a broader audience, I'd highly suggest that potentially complex terms be briefly explained within the text, in keeping with the authors' writing style. This will enhance readability;

We have added additional clarification on some key terms, including:

(a) RD prevalence (pg. 3, lines 45-47: “Its prevalence is estimated to be 5-10%, although the most common estimates are based largely on data from English-speaking countries [1-2]”)

(b) Rapid automatized naming/RAN (pg. 3, lines 54-55: “RAN refers to the speed with which one can name a series of presented stimuli,...”)

(c) Pseudowords (pg. 7, line 142): “fake but pronounceable ‘words’”.

4) Provide a well-structured presentation of the background, explaining the significance of your study and objetives;

A more comprehensive justification for the study, including its practical implications, has been added (see pg. 6, lines 122-132):

“Prior results have shown that abnormal low-γ phase-locking has been linked to deficits in phonological processing [11]; that those with RD show altered lateralization patterns [15-16, 24]; that reduced left-lateralized entrainment in the low-γ range is correlated with poorer phonological processing and rapid naming in RD [14]; and that there are developmental effects on low-γ power during typical development [19]. What is lacking in the literature is a clear consensus on resting-state dynamics in the developing reading and language network. Such a consensus allows for a better understanding of the development of children’s language and reading processes. Endogenous low-γ power, if associated with phonological processing, reading, and RD, would indicate both a potential biomarker and a more generalized processing deficit in the intrinsic auditory and language network in RD. Such a biomarker would suggest that the functional organization of the ‘RD brain’ diverges relatively early.”

We also add a more specific “Hypotheses” section immediately prior to “Materials and Methods” (pg 7, lines 133-147). The text is not new, but has been reorganized from the previous draft.

Methods

This section is very well-written and I'm quite happy with its rigour, clarity, and presentation. I have a few comments to consider:

1) This section needs primarily focus on the core aspects of the study. Information about Tables, sample characteristics, and other details may be more appropriately placed in the Results section;

We have moved the discussion of statistics related to rates of comorbidity, as well as Tables 1 and 2, to the beginning of the results section. The new section, entitled “Comorbidity rates and demographics”, begins on pg. 19, line 414. Additional discussion of these results has also been added (pgs. 20-21, lines 426-449):

“For the TR vs. RD-full comparisons, as expected, differences in diagnostic frequency emerged for ADHD, language disorder, and specific learning disorders of reading, where the RD-full group had significantly greater diagnostic rates than TRs (all ps <.05). Unexpectedly, the reverse effect was observed for Generalized Anxiety Disorder, where TRs were diagnosed at greater rates compared to RDs (p <.05). This could be due to the recruitment strategy used by CMI, which emphasizes community-referred recruitment of those with suspected neurodevelopmental and neuropsychiatric conditions as part of their initiative to build a robust database that “captures the broad range of heterogeneity and impairment” [25]. Thus, the RDs could have been recruited largely based on reading or other academic deficits, while TRs likely would have been recruited at higher rates for non-reading-related symptoms, such as anxiety. Additional descriptive information on the subsamples affected by the more common comorbidities, such as language disorder and anxiety disorders, are provided in Supporting Information (Tables S1-S3). For the RD-only vs. RD-ADHD comparisons, the RD-ADHD group had a self-evidently higher rate of ADHD diagnoses, while all other comparisons were non-significant (ps >.1).

“Table 2 describes demographic and behavioral variables, again for the N = 261 sample for the temporal lateralization analyses. The TRs and the RD-full group differed on SES [Mann-Whitney U = 11109.50 (p < .001), Cohen’s d = .380, 95% CIs (.250, .496)], with the typical readers coming from families with a higher annual income, consistent with prior literature on the relationship between SES and reading ability [37, 38]. The TRs also performed better (ps < .001) on both FSIQ [Cohen’s d = 1.221, 95% CIs (.949, 1.491)] and PIQ [Cohen’s d = .534, 95% CIs (.280, .787)], as well as – self-evidently – both TOWRE subtests [Cohen’s d = 3.587, 95% CIs (3.189, 3.983) and 3.494, 95% CIs (3.101, 3.883) for PDE and SWE, respectively]. Follow-up comparisons between the RD-only and RD-ADHD subgroups revealed no significant demographic or behavioral differences (all ps > .05). These results support the use of both WISC PIQ and SES as confounding variables for the analyses.”

--- Participants

2) The authors can consider clarifying who is allowed to use the CMI-HBN, offering more details on this aspect;

We have clarified on pgs. 7-8, lines 151-155 that the CMI neuroimaging/EEG data “is publicly available for download on the CMI HBN website (see Data Availability statement)”, while the behavioral and phenotypic data are “available to researchers upon completion of a Data Usage Agreement, to be signed by an authorized institutional official different from the principal investigator”.

3) Include the IRB number;

As all data is provided after being de-identified, with all protected health information removed, our study did not meet the criteria for human subjects research. This disclaimer is provided on pg. 8, lines 155-158: “They are provided in a de-identified/anonymized format such that none of the authors have access to any information which could be used to re-identify the participants. Therefore, additional participant consent or involvement was not required, and no further approval was sought from the University of Connecticut’s Institutional Review Board”.

4) Please provide a clear explanation for the chosen age range (13-), as it may not be immediately evident to all readers; Consider referencing Cellier's study from 2021, which discusses age and wave oscillations, although the age range differs slightly;

The provided paper by Cellier et al. (2021) is indeed interesting, suggesting that developmental effects on resting-state neural oscillations extend also to the alpha and theta bands. We provide justification for restricting the analyzed sample to ages 6-13 on pg. 8, lines 160-167: “...the age range for this study was restricted to those aged 6-13 years. Reasoning was threefold: (1) age 6 is when most children achieve early reading skills such that both reading metrics and individual differences are stable and reliable (standardized scores on tests of word reading may be derived starting at 6, and a majority of children with reading impairments are identified during this time frame), (2) within this age range, PA contributes to individual differences in word-reading and word-reading is reflective of decoding ability [8, 26], and (3) developmental changes in the brain possibly related to neural oscillations in the gamma range have been documented during this time frame [19]”.

5) Include Cronbach values for the WISC and TOWRE. They are known, but providing thiese values ensure alignment to best practices;

We have added the following information on reliability estimates for both WISC-V and TOWRE-2:

WISC-V (pg. 8, lines 168-169): “...[Wechsler Intelligence Scale for Children (WISC), internal consistency of primary index scores = .88-.93]...”

TOWRE-2 (pg. 8, line 173): “...(TOWRE-2, alternate-form reliability = .91-.95)...”

The appropriate references have also been added, both in-text and in the References section.

6) Explain why z-scores were not utilised;

For our study, we did not consider the conversion of continuous dependent or independent variables to z-scores to be necessary for our analyses for the following reasons: (1) the scores we included in the analyses had already been age-standardized; and (2) we did not create any composite scores for our analysis that were derived from multiple measures on different scales.

7) I've noticed something that can be misinterpreted or, if correctly interepreted, a drawback. I'd like the authors to clarify:

The criteria for the TR group, specifically using TOWRE scores >90, may not be the most suitable control for the RD group with TOWRE scores <85. The TR group is identified based on having TOWRE scores significantly higher than the RD group, but this may not adequately control for the presence of comorbidities or other factors that could influence the analysis of both groups.

The concern is not about using a non-restricted sample but rather the potential lack of a well-matched control. It's important to consider matching demographics or other variables that could make the TR and RD groups more similar for a more robust interpretation and explanation of the findings;

Thank you for addressing this topic. The effect of common comorbidities in various disorders deserves its own thorough discussion. We acknowledge that there is a tradeoff in our study between perfectly matched samples and a more ‘naturalistic’ approach that acknowledges the greater frequency of certain comorbidities in those with RD, such as ADHD, language disorders, and other specific learning disorders as seen in Table 1. We have therefore added additional supplementary information with data regarding differences in demographic variables and low-gamma power/lateralization due to the more common comorbidities of language disorder and anxiety, for which diagnostic rates differed between the TRs and RDs (Tables S1-S3). We also discuss differences in comorbidity rates more at the beginning of the Results section, with reference to the new Tables S1-S3 (pg. 20, lines 426-449); see cited text in the above comment on Methods 1).

Including comorbidities is a limitation of our study which we now discuss more extensively in the Discussion, in a new subsection on Implications and limitations (pg. 35, lines 741-): 

“The promise of analyzing endogenous rhythms within the brain is that we will one day be able to reliably identify biomarkers within clinical populations without the need for extensive task-based diagnostic batteries. Additionally, such biomarkers will grant us insight into the mechanisms underlying impairment in reading skills and any shared neural substrates with other well-known comorbid conditions. The current study, although its results are counterintuitive and have very small effects, are consistent with the position that there are possible functional differences between those with and without reading deficits during a passive EEG paradigm. However, any such differences must be met with a considerable degree of caution, given the small effects and the lack of replicability within this field. Given the current state of research, we do not advocate for the use of EEG biomarkers in any clinical setting for the identification of RD or dyslexia.

“One limitation of our study is that in the analyzed sample we had a considerable reduction in perfectly matched populations in favor of a more ‘naturalistic’ approach which allows for the occurrence of comorbid conditions (Table 1). While we tried to correct as much for spurious demographic variables as possible and note that rates of most comorbidities did not differ significantly between the TR and RD groups, there is an implicit degree of uncertainty in how the effects of comorbidities might manifest within each subgroup. This is especially true given our findings in the RD-ADHD subgroup, which were distinct from the RD-only subgroup in relation to temporal lateralization. Finally, it is possible that some of the control group participants may have missed receiving what would have been an appropriate RD diagnosis during childhood but had been able to compensate for any deficits (such that they tested within the normal range on TOWRE) by the time they were assessed. This is a concern that exists throughout reading research, particularly with adults, as it is difficult to objectively measure how much one struggled with reading during childhood retroactively. However, given that the current study was done largely with children under the age of 10, we believe that the chance that a high proportion of the controls had successfully compensated for any previous reading difficulties is relatively low.” 

We would also like to emphasize that most of the other non-language, non-ADHD comorbidities present in Table 1 are in relatively low frequencies and/or not significantly different in frequency between subgroups. On a related note, we were interested in isolating the unique effects of the most common comorbidity with RD - ADHD - due to its high co-occurrence in our sample and in RD more broadly (pgs. 10-11, lines 223-225): “Because we include common comorbidities, we performed exploratory/secondary analyses comparing RDs with and without the most common comorbidity, ADHD (accounting for 35.0% of the RD-full sample prior to quality control)”.

8) Please clarify the criteria and procedures used for achieving a consensus on diagnoses;

The consensus diagnoses are based on the internal procedure used for the CMI HBN project. This has been specified on pgs. 9-10, lines 193-204: 

“The consensus diagnoses are based on a combination of (1) psychiatric and medical history collected during a prescreening phone interview (a participant may have previously received an official diagnosis prior to involvement in the project); (2) clinician-administered diagnostic assessments and observations, including administration of the Kiddie Schedule for Affective Disorders and Schizophrenia (KSADS, Cronbach’s = .66, range: .25-.86) [34], which can be used to identify ADHD, major depressive disorder, generalized anxiety disorder, conduct disorders, and schizophrenia, among others; and (3) clinician-administered follow-up assessments, such as the Clinical Evaluation of Language Fundamentals (CELF-5, internal consistency of index scores = .92-.97) for identifying suspected language disorder [35], as well as other language-centric assessments such as TOWRE-2. For a full set of all assessments, see [25]. Official diagnoses are based on clinician assessment of a combination of results from the relevant tests and self-reported information from children, parents, and teachers.”

9) Explain how differences unrelated to reading may occur due to the lack of a matched TR group;

We have added the following to the Participants section, on pg. 10, lines 210-216: 

“We acknowledge that this decision constitutes a tradeoff between perfectly matched TR and RD samples and a more naturalistic approach that allows for greater neurodiversity in our sample. This could result in group differences being observed that are driven at least in part by the effects of elevated comorbidities in RD, particularly ADHD; on the other hand, allowing other comorbidities in both groups could also artificially ‘deflate’ true group differences. Demographic differences between groups will be controlled for by including them as confounding variables in our statistical analyses.”

10) Consider categorising confounding variables as moderators rather than covariates;

After reviewing the language used, we believe that it will be more accessible to our audience to refer to the covariates as ‘confounding variables’. These changes have been made throughout the manuscript.

11) Please provide details on the tools and criteria used for diagnosing ADHD and other conditions, along with their Cronbach values;

We have added the relevant reliability estimates (Cronbach’s alpha or other measures of internal consistency) where appropriate, along with their in-text citations and full citations in the References section; for details see our responses to Methods comments 5) and 8) above. 

12) Move the section explaining analyses to its proper location;

The details of planned statistical analyses are explained in the Analyses subsection of the Methods. Additionally, results of the preliminary demographic and behavioral analyses (including the accompanying Tables 1 and 2) have been moved to the beginning of the Results section (pg. 19, line 414), as requested.

13) Place tables and statistics (e.g. line 185) in their respective sections;

Please see our response to previous comment 12) above.

14) Please clarity why MI is preferable to maximum likelihood estimation in this context, potentially within the analyses section;

There have been some comparisons done between these two methods (see references below) or between MI and other methods using reading profile data (Eckert et al., 2018), which constitute important methodological contributions to the field. Ultimately, the decision was made largely based on its validation in Eckert et al. (2018) and due to the available implemented and validated methods in SPSS, which utilizes MI based on linear regression or predictive mean matching (with the latter being used in Eckert et al.). We added the following sentence to pg. 11, lines 239-240: “The MI method comes pre-implemented in SPSS, with various options for model types for scale variables (linear regression or predictive mean matching)”.

Eckert MA, Vaden KI Jr, Gebregziabher M; Dyslexia Data Consortium. Reading Profiles in Multi-Site Data With Missingness. Front Psychol. 2018 May 8;9:644. doi: 10.3389/fpsyg.2018.00644. PMID: 29867632; PMCID: PMC5952106.

Shin T, Davison ML, Long JD. Maximum likelihood versus multiple imputation for missing data in small longitudinal samples with nonnormality. Psychol Methods. 2017 Sep;22(3):426-449. doi: 10.1037/met0000094. Epub 2016 Oct 6. PMID: 27709974.

Ferro MA. Missing data in longitudinal studies: cross-sectional multiple imputation provides similar estimates to full-information maximum likelihood. Ann Epidemiol. 2014 Jan;24(1):75-7. doi: 10.1016/j.annepidem.2013.10.007. Epub 2013 Oct 18. PMID: 24210708.

15) Please include additional demographic details, such as gender proportions, IQ values and proportions (i.e. I've only seen IQ), and whether participants' family members or parents had a history of similar conditions;

Demographic details of sex, WISC full-scale IQ (FSIQ), dominant hand (Handedness), and socioeconomic status (SES) are provided in Table 2 (pgs. 21-22, line 450) and discussed in more detail in the Results section.

16) Consider adding a flowchart illustrating, because it's *really/ important, explaining the progression from the beginning, including sample selection, exclusions, and criteria;

A flowchart detailing the process of selecting the sample from CMI HBN, preprocessing, and the various criteria applied to get the final analyzed samples has been added (now Fig. 2) with a caption placed on pg. 14, line 315 (“Flowchart detailing all steps of EEG preprocessing and exclusionary criteria”).

17) Clarify how the sample size was reduced from 315 to the final observed number;

The following clarification was added to pg. 14, lines 309-311 regarding how N = 315 was reduced to the final sample on which outlier analyses were conducted: “Finally, the quality control and minimum-epoch criteria were applied, excluding an additional 53 participants (N = 262). A profile of these excluded participants is provided in Supporting Information (Table S4)”. Also see the flowchart referenced in response to previous comment 16) above.

18) Please address the issue of clinical diagnoses and their potential influence on the generalisability of findings;

We hope our responses to Methods comments 7) and 9) above sufficiently address the issues regarding comorbidities.

19) Please specify the sampling method;

We hope that our more detailed explanation regarding the selection of participants from the CMI dataset and new Fig. 2 provides a satisfactory explanation of participant sampling.

20) Please consider to xplain all variables measured and observed in the study;

We have added text throughout to ensure that all variables are thoroughly explained, in particular all assessments used (WISC, TOWRE and its subtests, etc.), field-specific acronyms (such as RAN), specific constructs or abilities (e.g. phonemes, phonological awareness, phonemic processing/decoding, pseudowords), as well as definitions for specific covariates where appropriate (e.g., how socioeconomic status was measured).

--- EGG

1) Please describe how instructions and the conditions were made uniform for all participants;

All participants were given the same single EEG paradigm, which consisted of the same visual fixation cross stimuli and recorded voice prompts as described in the “Resting-state paradigm” section (pg. 12, line 250). We have also made an improved version of Fig. 1 which we believe better communicates the paradigm used for all participants.

2) Clarify the approach to managing challenges in resting state EEG, especially with regard to the presence of alpha waves. As the authors are aware, resting state is where the alpha likes to play. I'd like authors to clarify this, especially when avoiding that some waves bury in others;

We have added the following discussion to the “Implications and limitations” subsection (pg. 36, lines 766-783): 

“A second limitation are the challenges involved in analyzing resting-state EEG data. There are a few advantages to using resting-state EEG, chief among which are its relative ease of data collection and cross-study comparability relative to task-based designs. Resting-state EEG also has value for characterizing intrinsic functional activity in the brain. At the same time, there are several challenges involved in analyzing and interpreting this data. The first is that in the realm of spectral analysis of EEG data certain frequency bands – such as alpha – are more commonly studied during resting-state, while the gamma band is relatively less well-studied [66]. The consistency of these results may vary as a function of whether the eyes are kept open or closed: an extensive review of spectral analyses for resting-state EEG in various disorders reported that results from EC studies have greater aggregated consistency than those of EO studies [66]. In this study we attempted to elucidate an ‘average’ resting-state gamma profile derived from both EO and EC data, however future studies would do well to examine both states separately with sufficient data for both cases. There is also a degree of ‘fuzziness’ or overlap between functional properties ascribed to these different frequency bands: alpha has been reported to be associated with both performance on various intelligence measures and attention [72, 73], while higher-range gamma activity has also been proposed to be related to attention [64, 67], with alpha-gamma coupling being proposed to reflect early perceptual processing [73-75]. These processes are likely to be relevant not just for attention, but for the processing of visual information such as written language.”

3) Considering this, did the authors check the temporal pattern of the waves? Or SNS to avoid buried waves and also to check and minimise individual's differences, which is common in resting state;

We did not calculate the signal to noise-subtracted amplitudes on our EEG data as part of our analyses. Can the editor clarify whether this is what they are referring to by ‘SNS’?

4) Discuss the use of ICA and cross-modal aspects, or clarify when these will be reported;

We apologize, as we may have misunderstood this question. Can the editor clarify what they are referring to regarding ‘cross-modal aspects’ of ICA?

5) Explain how montages were applied and whether electrodes were pre-prepared for placement;

We have added additional information to the “Acquisition” subsection, which now reads as follows (pgs. 12-13, lines 263-271):

“Data were acquired from a high-density 128-channel net using the EGI (Electrical Geodesics, Inc.) Geodesic Hydrocel system. Collection was performed in a sound-attenuated room with a sampling rate of 500 Hz and on-line 0.1-100 Hz bandpass filter. The recording reference was Cz. Electrodes were pre-prepared before placement. Electrode impedance was checked prior to recording and kept below 40 kOhm throughout, being tested every 30 minutes; saline was added if needed [25]. Simultaneous eye tracking was performed by recording eye position and pupil dilation with an infrared video-based eye tracker at a sampling rate of 120 Hz. The 9 EOG channels were placed on the forehead, outer and inner canthi (electrodes E8, E14, E17, E21, E25, E125, E126, E127, and E128).”

6) Detail the rationale for using the PREP - as the use of Python for preprocessing is being evident. Actually, this is just a matter of curiosity;

PREP was used as part of the GUI-based Automagic MATLAB toolbox for EEG preprocessing (see reference below). Automagic provides access to a large number of options for large-scale preprocessing and efficient visualization of data quality metrics across all participants, along with automated quality classification. We therefore found it intuitive to use Automagic for the majority of the preprocessing, while using MNE for additional derivation of features for analysis and visualization.

Pedroni A, Bahreini A, Langer N. Automagic: Standardized preprocessing of big EEG data. Neuroimage. 2019 Oct 15;200:460-473. doi: 10.1016/j.neuroimage.2019.06.046. Epub 2019 Jun 21. PMID: 31233907.

7) I am relieved because the authors used a higher-density, so it's a little bit 'easier' to cut artifacts. The montage was set before or just a little bit beforing resting state?;

The montage was set before resting-state, as resting-state data was collected during a larger paradigm consisting of multiple active and passive paradigms - see Alexander et al., citation [25] for further details:

Alexander LM, Escalera J, Ai L, Andreotti C, Febre K, Mangone A, et al. An open resource for transdiagnostic research in pediatric mental health and learning disorders. Sci Data. 2017 Dec 19;4:170181. doi: 10.1038/sdata.2017.181. PMID: 29257126; PMCID: PMC5735921.

8) Please clarify the use of a 53 Hz (i.e. notch filter?) and debate epochs (i.e. or why it was divided into two sections);

We clarify that a notch filter was not used on pg. 13, line 280. We have also reorganized the Preprocessing and quality control sections in the following manner: the 2 sections have been combined (see pg. 13, line 272-), with the 1st paragraph briefly summarizing the overall process (also see the new Fig. 2); the 2nd paragraph discusses channel flagging and quality criteria in more detail; finally the 3rd paragraph discusses how the preprocessed data were imported into MNE and re-epoched, along with specific criteria used, and concluding with an explanation of the final sample.

9) A fluxogram - a sequential fluxogram - would be fantastic here. This explanation for EEG is commendable;

Thank you for your compliment. We have added an additional flowchart describing EEG preprocessing and participant sampling (now Fig. 2), and hope that this clears up any ambiguity.

10) Avoid beginning sentences with abbreviations (e.g. line 227);

We have revised this sentence as follows (pg. 13, lines 270-271): “The 9 EOG channels were placed on the forehead, outer and inner canthi (electrodes E8, E14, E17, E21, E25, E125, E126, E127, and E128)”.

11) Clarify whether spectral and ERP analyses were conducted;

For our study, we conducted only EEG spectral analyses, as we analyzed the power spectrum and particularly the low-gamma band using estimates of power spectral density (PSD). ERP-type analyses were not performed, as we did not analyze the averaged signal across epochs.

12) Explain the criteria for identifying individual artifacts, particularly without EMG. The authors checked it manually too?;

We believe that “EMG” is in reference to electromyography which is used to measure the electrical activity of muscles. These types of artifacts are presumed to be captured with the use of independent component analysis (ICA) by ICLabel, which allows the user to select which artifacts to exclude such as those from the heart, muscle, and eyes; as well as options for line noise and channel noise. Due to the large sample used for this study, we utilized Automagic’s feature of automatic quality ratings based on chosen quality criteria, which provide a high-level estimate for the degree of artifact contamination; see also our response to comment 13) below. Details are provided on pg. 13, lines 278-279: “ICA was done with ICLabel [42]. Artifacts associated with muscle, heart, and eye activity were removed”.

13) Channels flagging need to have the limits presented;

We have added additional details regarding the criteria for channel/timepoint flagging (pgs. 13-14, lines 285-297): 

“PREP flags channels based on the following criteria: high amplitudes with a Z-score >5; a maximum correlation <.4 with any other channel (within a time window of 1 s); predicted channel activity [based on a random 25% of (so far) ‘good’ channels] that has a correlation <.75 with the true channel activity, within a certain fraction (>.4) of non-overlapping 5 s time windows; and unusually high-frequency noise, indicated by the ratio between the median absolute deviations of the high-frequency signal (>50 Hz) versus the low-frequency signal having a Z-score >5 [36]. Residual bad channel detection was run after the entire preprocessing pipeline finished to catch any remaining channels with excessively high or low variance. A participant’s EEG data was also flagged based on a ratio of high-variance time points exceeding .20, and individual channels were flagged based on overall high amplitude. Criteria for excessively high/low variance were voltage thresholds defined as a function of the standard deviation of the voltage measures across all time points or channels, 25 × SD(mV). The criterion for overall high amplitude was a 40-mV threshold for absolute voltage magnitude.”

14) Please clarify the spherical method and its potential influence on data;

We add a brief description of the spherical method on pg. 14, lines 297-300: 

“Bad channels were interpolated using the spherical method, which involves projecting all channels to a unit sphere and mapping the identified ‘bad’ channels to surrounding good channels to estimate missing EEG data [43]. This is one of the most popular methods for bad-channel interpolation, although it may perform less well if there are many bad channels adjacent to one another.”

15) Define the criteria for 1-second epochs;

We have added the following information to pg. 14, lines 306-307: “Standards for epoch rejection were a maximum peak-to-peak signal amplitude (PTP) of 100 μV, and a minimum PTP of 1 μV.”

16) ust now the authors mention that partiipanst were removed - but why? Maybe the data would be inflated or diferent with them?;

In case the editor is referring to exclusion criteria, we clarify that an additional 76 participants were removed due to either missing 1 TOWRE subtest score once the data was examined for further analysis (N = 23; during the initial collection of participant data, we failed to check whether both TOWRE subtest scores were present rather than at least 1) or due to failing quality control minimum ‘good’ epoch criteria as described previously (N = 53). We hope that the new Fig. 2 we added helps to clarify this process, and apologize for any ambiguity. 

The classification of participants with missing TOWRE subtests into reading groups would be subject to greater error, as some children may have below-average decoding ability but above-average word-reading, which we would not be able to determine from this sample with missing subtest scores. Including those with poor quality EEG data would similarly introduce error into our analysis, as results derived from these data are less reliable.

17) The exclusion of values that exceed certain expected or the mean, is indeed an interesting step in the data preprocessing. However, researchers might want to know more about the specific criteria used for this exclusion. Since your study has a substantial sample, some might argue that the exclusion of these values may not significantly influence the overall finding. Providing further details or explanation for these exclusions could help address potential questions or concerns regarding this aspect of your data processing;

Regarding the outlier exclusion, we include the following explanation in the ‘Outliers’ subsection of the Methods (pg. 16, lines 341-343): “Outliers were determined to fall outside the range [1st quartile - (1.5 × IQR), 3rd quartile + (1.5 × IQR)], where IQR is the interquartile range, therefore being much higher or lower than the rest of the data”.

We considered that it may also be helpful to visualize our examination of DVs. We add the following to pg. 16, line 345: “Q-Q plots are presented in Supporting Information (Fig. S1)”.

--- Calculation for low-gamma

1) Clarify whether you calculated PSD for each epoch or used a different approach. For example, I got the a'll epochs', but defining a range would be also useful for a better control;

PSD was calculated for each epoch separately and then averaged across all epochs. To clarify this in the paper, the paragraph on pg. 15, lines 328-336 has been rewritten as follows:

“For each channel, power spectral density (PSD) was estimated using the multitaper method as implemented in the MNE-Python package. The PSD was used to calculate mean gamma band-power in the 30-45 Hz range for all epochs separately; gamma band-power was then averaged across all epochs; and this average value was finally subjected to a log transform of 10 * log10(x) to derive absolute power in units of μV2/Hz (dB). The total number of epochs per participant ranged from 100-300 (median = 225 epochs) and did not differ as a function of RD status (t-test, p = .664) or ADHD status within RDs (p = .903). The average channel values were then additionally averaged across all channels within the left and right hemisphere temporal lobes (separately for each hemisphere).”

2) Explain the rationale for choosing 45 Hz. This is a matter of curiosity. If you have the opportunity to check Haenschel's study (2000) there's explanation of this question and the one about buried waves;

Our rationale is that we were particularly interested in focusing our analysis on the low-gamma range that corresponds to the tracking of phonemic-rate auditory stimuli in speech. It is true that the definition of the gamma range may vary considerably based on the functional properties associated with it. For instance, high-frequency gamma oscillations >45 Hz are active in high-level cognitive and attentional processes and could be atypical in ADHD (Başar et al., 2000; Ray et al., 2008; Rudo-Hutt, 2015), but as this was not our primary objective we consider it out of scope for this study. We did add the following sentence at the end of the section describing the calculation of low-gamma power (pg. 15, lines 336-338): “The rationale for choosing the range 30-45 Hz was based on previous findings of RD abnormalities in low-gamma frequencies within this range, and to therefore remove irrelevant and potentially spurious higher-frequency gamma rhythms [12, 18]”.

Başar, E., Başar-Eroğlu, C., Karakaş, S., & Schürmann, M. (2000). Brain oscillations in perception and memory. International journal of psychophysiology: official journal of the International Organization of Psychophysiology, 35(2-3), 95–124. https://doi.org/10.1016/s0167- 8760(99)00047-1

Ray, S., Niebur, E., Hsiao, S. S., Sinai, A., & Crone, N. E. (2008). High-frequency gamma activity (80- 150Hz) is increased in human cortex during selective attention. Clinical neurophysiology: official journal of the International Federation of Clinical Neurophysiology, 119(1), 116–133. https://doi.org/10.1016/j.clinph.2007.09.136

Rudo-Hutt, A. S. (2015). Electroencephalography and externalizing behavior: a meta-analysis. Biological psychology, 105, 1–19. https://doi.org/10.1016/j.biopsycho.2014.12.005

3) Confirm if MNE automatically used FFT as the algorithm for calculations;

MNE’s function for calculating PSD using the multitaper method provides information on how the EEG signal power is distributed across frequencies. This method does not rely on the FFT algorithm.

4) Please clarify whether any manual data checking was conducted at each step of the analysis;

Manual data checking was done prior to the main analyses for the purpose of normality checking and outlier exclusion (see Fig. S1).

--- Bayes

1) If the authors find that the explanation of BF₀ and BF₁₀ is a central focus of their study, they can mainting it. Otherwise, provide a concise explanation or refer readers to relevant sources for more in-depth information. This approach will help maintain the focus on the main research objectives and findings while still providing some context for the Bayesian analyses;

Thank you for this suggestion. We have shortened our explanation of Bayes factors (in particular removing unnecessary equations), which now reads as follows (pg. 17, lines 364-374): 

“The Bayes factor (BF10) is the ratio between the probabilities of observing the data given that the alternative hypothesis (H1) is true, and observing the data given that the null hypothesis (H0) is true. A BF10 < .300 indicates moderate evidence favoring the null hypothesis, while a BF10 > 3.000 indicates moderate evidence in favor of the alternative hypothesis; values in between indicate that evidence is either anecdotal (.300 < BF10 < 3.000) or absent, BF10 ≈ 1 or log(BF10) ≈ 0 [45]. The inclusion Bayes factor (BFincl) is also calculated for all individual factors in each model to determine the contribution of individual effects. The BFincl estimates an effect’s unique contribution by comparing all models that contain the effect to equivalent models without that effect.”

2) Consider using an exponential scale for Bayes factors (BF) to enhance the rigour;

We have made appropriate edits to our in-text explanations and figures which include BF information (in particular see updated Figs 5, 6), now presenting these values as log(BF). With this change, BFs <1 (in greater support of the null hypothesis) are presented as negative, which we believe provides greater clarity for interpretation (pgs. 17-18, lines 385-387): “Where discussed, Bayes factors are also presented in the form log(BF10) for clarity; positive values indicate greater evidence in favor of the experimental hypothesis, while negative values indicate evidence in favor of the null”.

3) Please explain if you explored different models to assess their usability and to support the choice made;

We test and report only the models which were planned prior to undertaking the data analysis in pursuit of our research questions (as well as any post-hoc tests which emerged from interactions or ambiguous results).

--- Analyses

1) Please clearly state the software used for the analyses. Include effect sizes and CIS, and maintain consistency in the number of decimal places. This will ensure the same rigour;

Software versions used for analysis have been made more explicit, at the beginning of the “Analysis plan” subsection (pg. 17, lines 380-382): “All preliminary statistical checks (Box’s test and Levene’s test) and tests of demographics/behavioral data were performed in SPSS v28.0.0.0. All Bayesian analyses were performed in JASP v0.14.1”.

Bayesian measures of effect size are added where previously missing. Values for 95% credible intervals are provided either in-text or in figure captions as needed (pg. 18, lines 387-389): “Bayesian estimates of effect size are given as either model averaged posterior distribution values (δ, for ANCOVA/t-test) or posterior coefficients (β, for regression), with 95% credible intervals (CIs)”.

2) Consider you have a complex data and, consequently, analyses. Using Bayesian for various types of models can indeed help mitigate issues related to multiple analyses and provide a more unified approach. Bayesian offer advantages in handling uncertainty, which can be especially valuable when dealing with complex, multiple data and multiple DVs;

In Bayesian regression, oauthors can observe posterior distribution for model parameters and make probabilistic statements about the relationships between variables. Additionally, Bayesian SEM allows for more flexible modeling of complex relationships among variables, making it suitable for your data. And, important as well, Bayesian MANOVA provides a comprehensive way to analyse data when there are more than one DVs, making it a suitable alternative to multilevel analyses;

Thank you for your rigorous attention to detail regarding our paper and analyses. The various applications of Bayesian methods for statistical analysis are indeed promising, particularly for complex data, and we are optimistic that other researchers in our field will utilize them more in the coming years.

Results

Please use scatter and heatmaps whenever possible, aligning with the rigour of your study.

We have added additional descriptive group-level EEG topographic plots for gamma power to the revised Fig. 4, and provide a scatterplot of decoding ability vs. lateralization in Fig. 7.

Reviewer 1

The findings are good and writing style is also appreciated. However, too much of variety in the sample types with unequal sample sizes could have potentially led to inconclusive results. Focussing on few common populations where reading disability is encountered may have provided better insights into their neural processing of reading.

We thank the reviewer for their compliments. We agree that the issue stated is a significant limitation to our study, and have made some additions to both the Results and the Discussion sections, as well as the Supplementary Information, to reflect this. In particular we added additional information regarding differences in demographic variables and low-gamma power/lateralization due to the more common comorbidities of language disorder and anxiety, for which diagnostic rates differed between the TRs and RDs (Tables S1-S3). We also discuss differences in comorbidity rates more at the beginning of the Results section, with reference to the new Tables S1-S3 (pgs. 20-21, lines 426-449):

“For the TR vs. RD-full comparisons, as expected, differences in diagnostic frequency emerged for ADHD, language disorder, and specific learning disorders of reading, where the RD-full group had significantly greater diagnostic rates than TRs (all ps <.05). Unexpectedly, the reverse effect was observed for Generalized Anxiety Disorder, where TRs were diagnosed at greater rates compared to RDs (p <.05). This could be due to the recruitment strategy used by CMI, which emphasizes community-referred recruitment of those with suspected neurodevelopmental and neuropsychiatric conditions as part of their initiative to build a robust database that “captures the broad range of heterogeneity and impairment” [25]. Thus, the RDs could have been recruited largely based on reading or other academic deficits, while TRs likely would have been recruited at higher rates for non-reading-related symptoms, such as anxiety. Additional descriptive information on the subsamples affected by the more common comorbidities, such as language disorder and anxiety disorders, are provided in Supporting Information (Tables S1-S3). For the RD-only vs. RD-ADHD comparisons, the RD-ADHD group had a self-evidently higher rate of ADHD diagnoses, while all other comparisons were non-significant (ps >.1).

“Table 2 describes demographic and behavioral variables, again for the N = 261 sample for the temporal lateralization analyses. The TRs and the RD-full group differed on SES [Mann-Whitney U = 11109.50 (p < .001), Cohen’s d = .380, 95% CIs (.250, .496)], with the typical readers coming from families with a higher annual income, consistent with prior literature on the relationship between SES and reading ability [37, 38]. The TRs also performed better (ps < .001) on both FSIQ [Cohen’s d = 1.221, 95% CIs (.949, 1.491)] and PIQ [Cohen’s d = .534, 95% CIs (.280, .787)], as well as – self-evidently – both TOWRE subtests [Cohen’s d = 3.587, 95% CIs (3.189, 3.983) and 3.494, 95% CIs (3.101, 3.883) for PDE and SWE, respectively]. Follow-up comparisons between the RD-only and RD-ADHD subgroups revealed no significant demographic or behavioral differences (all ps > .05). These results support the use of both WISC PIQ and SES as confounding variables for the analyses.”

Also see the following newly added section in the Discussion (pg. 35, line 741-, “Implications and limitations”):

“The promise of analyzing endogenous rhythms within the brain is that we will one day be able to reliably identify biomarkers within clinical populations without the need for extensive task-based diagnostic batteries. Additionally, such biomarkers will grant us insight into the mechanisms underlying impairment in reading skills and any shared neural substrates with other well-known comorbid conditions. The current study, although its results are counterintuitive and have very small effects, are consistent with the position that there are possible functional differences between those with and without reading deficits during a passive EEG paradigm. However, any such differences must be met with a considerable degree of caution, given the small effects and the lack of replicability within this field. Given the current state of research, we do not advocate for the use of EEG biomarkers in any clinical setting for the identification of RD or dyslexia.

“One limitation of our study is that in the analyzed sample we had a considerable reduction in perfectly matched populations in favor of a more ‘naturalistic’ approach which allows for the occurrence of comorbid conditions (Table 1). While we tried to correct as much for spurious demographic variables as possible and note that rates of most comorbidities did not differ significantly between the TR and RD groups, there is an implicit degree of uncertainty in how the effects of comorbidities might manifest within each subgroup. This is especially true given our findings in the RD-ADHD subgroup, which were distinct from the RD-only subgroup in relation to temporal lateralization. Finally, it is possible that some of the control group participants may have missed receiving what would have been an appropriate RD diagnosis during childhood but had been able to compensate for any deficits (such that they tested within the normal range on TOWRE) by the time they were assessed. This is a concern that exists throughout reading research, particularly with adults, as it is difficult to objectively measure how much one struggled with reading during childhood retroactively. However, given that the current study was done largely with children under the age of 10, we believe that the chance that a high proportion of the controls had successfully compensated for any previous reading difficulties is relatively low.”

Reviewer 2

1. Introduction (page 3, lines 43-44): It would be beneficial to clarify what exactly “general population” indicates. Is it referring to the global population or to a specific country or region?

We agree some elaboration would be helpful. In our usage, ‘general population’ refers to an estimate of overall prevalence regardless of other factors (such as those with other diagnoses that are often comorbid with dyslexia, or those who have family history). We have removed the clause “in the general population” since simply using the word “prevalence” conveys the intended message, and we have also added a clause explaining that this is largely based on data from English-speaking countries (pg. 3, lines 45-47): “Its prevalence is estimated to be 5-10%, although the most common estimates are based largely on data from English-speaking countries [1-2]”.

2. Introduction (page 3, line 46): It would be better to provide a few examples of neurodevelopmental disorders here.

We have changed the indicated sentence to the following to better reflect the content of the cited references (pg. 3, lines 49-51): “RD is often comorbid with other specific learning disorders, attention deficit hyperactivity disorder (ADHD), language disorders, and anxiety/depression [5-7].”

3. Introduction (page 3, lines 48-49): To accommodate potential readers from different areas, it is suggested to provide brief explanations or definitions for certain terminologies, such as “rapid automatized naming.”

The following explanation for RAN has been added (pg. 3, lines 54-55): “RAN refers to the speed with which one can name a series of presented stimuli”

4. Introduction (page 5-6, lines 107-111): Please provide a comprehensive and in-depth discussion of the potential clinical or practical implications that can be derived from this study.

A more comprehensive justification for the study, including its practical implications, has been added to the revised and reorganized Introduction. This is based in part on reorganization of the Introduction. Specifically see pg. 6, lines 120-132: 

“The aim of the current study is to investigate low-γ power and lateralization at rest in children with and without RD in an age range where children are learning to read (6 to 13 years), expanding prior resting-state studies in beginning readers and adolescents. Prior results have shown that abnormal low-γ phase-locking has been linked to deficits in phonological processing [11]; that those with RD show altered lateralization patterns [15-16, 24]; that reduced left-lateralized entrainment in the low-γ range is correlated with poorer phonological processing and rapid naming in RD [14]; and that there are developmental effects on low-γ power during typical development [19]. What is lacking in the literature is a clear consensus on resting-state dynamics in the developing reading and language network. Such a consensus allows for a better understanding of the development of children’s language and reading processes. Endogenous low-γ power, if associated with phonological processing, reading, and RD, would indicate both a potential biomarker and a more generalized processing deficit in the intrinsic auditory and language network in RD. Such a biomarker would suggest that the functional organization of the ‘RD brain’ diverges relatively early.”

5. Materials and Methods – Participants (page 7, lines 138-139): Please add a reference supporting this statement.

The relevant references were added to the indicated sentence (pg. 8, line 161).

6. Materials and Methods – Participants (page 7, lines 145-146): Please add the considerations for why a cutoff of 70 was chosen for the IQ scale, as opposed to other values.

A brief explanation of our reasoning for the chosen cutoff was added: “to ensure that we had a representative sample population with a normal distribution, that the participants were cognitively able to complete the assessments reliably, and that reading deficits observed in our RD sample were not attributable to intellectual disability” (pg. 8, lines 169-172).

7. Materials and Methods – Participants (page 8, lines 156-157): Is it possible that not receiving a prior clinical diagnosis of RD was simply because the individual did not go to a doctor, which doesn't mean the subject doesn't have RD?

This is a valuable question, and an important consideration for reading research. We have attempted to control for this possibility in our classification criteria. The way we do this for the controls is by requiring not only that they not have a diagnosis, but they must also have sufficiently high TOWRE scores (pg. 9, lines 183-184): “(2) a TR group who (a) had not received a prior clinician diagnosis of RD and (b) had both TOWRE subtest scores >90”. 

Of course, it is still possible that a child could have both (a) missed receiving a diagnosis at a young age because they did not go see a clinician, and (b) compensated for these deficits, such that they now test within the normal range on TOWRE. We have added an acknowledgement of this possibility in our new section on Implications (pgs. 35-36, lines 758-765): 

“Finally, it is possible that some of the control group participants may have missed receiving what would have been an appropriate RD diagnosis during childhood but had been able to compensate for any deficits (such that they tested within the normal range on TOWRE) by the time they were assessed. This is a concern that exists throughout reading research, particularly with adults, as it is difficult to objectively measure how much one struggled with reading during childhood retroactively. However, given that the current study was done largely with children under the age of 10, we believe that the chance that a high proportion of the controls had successfully compensated for any previous reading difficulties is relatively low.”

8. Materials and Methods – Participants (page 8, lines 157-159): As the inclusion of comorbidities could introduce serious biases into the results, one of the major concerns of this study was whether any efforts were made to prevent or decrease their potential influences.

The effect of comorbidities on how disorders are expressed is a topic deserving of thorough analysis in its own right. We attempt to control somewhat for the effects of the most common comorbidity, ADHD (pgs. 10-11, lines 223-225): “Because we include common comorbidities, we performed exploratory/secondary analyses comparing RDs with and without the most common comorbidity, ADHD (accounting for 35.0% of the RD-full sample prior to quality control)”.

However, the inclusion of comorbidities is indeed one of the limitations of our study which we now acknowledge more explicitly in the Methods section discussing participants (pg. 10, lines 210-216):

“We acknowledge that this decision constitutes a tradeoff between perfectly matched TR and RD samples and a more naturalistic approach that allows for greater neurodiversity in our sample. This could result in group differences being observed that are driven at least in part by the effects of elevated comorbidities in RD, particularly ADHD; on the other hand, allowing other comorbidities in both groups could also artificially ‘deflate’ true group differences. Demographic differences between groups will be controlled for by including them as confounding variables in our statistical analyses.”

We also add a brief discussion of this issue in the Results (pgs. 20-21, lines 426-439): 

“For the TR vs. RD-full comparisons, as expected, differences in diagnostic frequency emerged for ADHD, language disorder, and specific learning disorders of reading, where the RD-full group had significantly greater diagnostic rates than TRs (all ps <.05). Unexpectedly, the reverse effect was observed for Generalized Anxiety Disorder, where TRs were diagnosed at greater rates compared to RDs (p <.05). This could be due to the recruitment strategy used by CMI, which emphasizes community-referred recruitment of those with suspected neurodevelopmental and neuropsychiatric conditions as part of their initiative to build a robust database that “captures the broad range of heterogeneity and impairment” [25]. Thus, the RDs could have been recruited largely based on reading or other academic deficits, while TRs likely would have been recruited at higher rates for non-reading-related symptoms, such as anxiety. Additional descriptive information on the subsamples affected by the more common comorbidities, such as language disorder and anxiety disorders, are provided in Supporting Information (Tables S1-S3). For the RD-only vs. RD-ADHD comparisons, the RD-ADHD group had a self-evidently higher rate of ADHD diagnoses, while all other comparisons were non-significant (ps >.1).”

We also discuss this issue in the new section on Implications and limitations (pg. 35, lines 752-758): 

“One limitation of our study is that in the analyzed sample we had a considerable reduction in perfectly matched populations in favor of a more ‘naturalistic’ approach which allows for the occurrence of comorbid conditions (Table 1). While we tried to correct as much for spurious demographic variables as possible and note that rates of most comorbidities did not differ significantly between the TR and RD groups, there is an implicit degree of uncertainty in how the effects of comorbidities might manifest within each subgroup. This is especially true given our findings in the RD-ADHD subgroup, which were distinct from the RD-only subgroup in relation to temporal lateralization.”

9. Materials and Methods – Participants (page 8, lines 163-166): Although comorbidities were allowed based on consensus diagnoses from clinicians, it is still unclear what considerations led to this decision (i.e., whether a comorbidity was allowed or not).

We have added the following (pg. 10, lines 205-210) to clarify this: “All comorbidities were permitted to be included, with the exception of severe intellectual disability due to our IQ cutoff criteria. Comorbid diagnosis of ADHD was also restricted to the RD population on account of its common association with RD [6]. Children with low-average TOWRE scores (85-90) were not included, as we wanted to ensure the presence of a gap in reading ability between the two main groups. After applying our exclusion criteria, some comorbidities were not represented in our sample (for instance, conduct disorders; see Results)”.

10. Materials and Methods – Participants (page 8, lines 167 and 170): It would be beneficial to include a profile of background characteristics of the 54 excluded subjects (i.e., 315 – 261) and to assess whether they differ from the included sample in some aspect.

The 53 participants whose EEG data did not meet the quality threshold have additional background information provided in Supplementary Information (new Table S4), including demographic variables and group designation. Differences between this sample and the N = 262 analyzable participants are also provided (pg. 14, lines 310-311): “A profile of these excluded participants is provided in Supporting Information (Table S4)”.

11. Materials and Methods – Participants (page 8, lines 174-175): It is encouraged to conduct supplementary analyses comparing individuals with RDs, both with and without second most common comorbidity as well. Even if limited to descriptive statistics, this would provide a more comprehensive picture.

The second most common (non-RD) comorbidity is language disorder or - if collapsing across all disorder subtypes - anxiety disorders. Additional descriptive information on these subsamples is given in the Supplementary Information Tables S1-S3 (pgs. 48-51).

12. Materials and Methods – Participants (page 8, lines 174-175): Please strengthen the implications regarding the comparison between RDs with and without ADHD in terms of clinical and practical aspects.

We added the following section to the end of this paragraph (pg. 11, lines 225-230): “It is of interest whether reading deficits manifest differently as a function of attentional deficits, both in regards to behavioral phenotypes and endophenotypes (e.g., EEG profiles). Deficits in attention may affect academic performance across multiple domains, including reading, and some have even proposed that comorbid RD and ADHD constitute a unique ‘subtype’ of reading disorder that is attributable to attentional deficits [6]”.

13. Materials and Methods – Participants (page 10, lines 185-187): Please discuss this unexpected reverse effect and its potential effects on the results of this study.

As part of our reorganization, we have moved the section where these results are reported to the beginning of the Results, in the subsection entitled “Comorbidity rates and demographics”. We now include the following discussion (pgs. 20-21, lines 428-437): “Unexpectedly, the reverse effect was observed for Generalized Anxiety Disorder, where TRs were diagnosed at greater rates compared to RDs (p <.05). This could be due to the recruitment strategy used by CMI, which emphasizes community-referred recruitment of those with suspected neurodevelopmental and neuropsychiatric conditions as part of their initiative to build a robust database that “captures the broad range of heterogeneity and impairment” [25]. Thus, the RDs could have been recruited largely based on reading or other academic deficits, while TRs likely would have been recruited at higher rates for non-reading-related symptoms, such as anxiety. In relation to its potential effect on the results, see additional descriptive information on the subsamples affected by the more common comorbidities, such as language disorder and anxiety disorders, are provided in Supporting Information (Tables S1-S3)”.

14. Materials and Methods – Participants (page 10, lines 189-190): Please explain the approach for testing non-random missingness of data.

Thank you for bringing this to our attention. We believe that this phrase was mistakenly used; the justification for using multiple imputation is to preserve statistical power (as stated), while preserving the statistical relationship between the imputed variable and variables used for imputation. For instance, SES is correlated with FSIQ and reading ability in this sample, so using a naive approach of imputing based on the median would artificially deflate these correlations. The phrase “non-random missingness”, which refers to the case where data is more likely to be missing as a function of some variable, has been removed, as it is not true in our sample.

15. Materials and Methods – Quality control (page 13, lines 238-239): It could still be unclear for readers how to define excessively high or low variance for channels. It would be beneficial to provide an instance.

We have provided the following additional information regarding our criteria for channel flagging (pgs. 13-14, lines 285-297): 

“PREP flags channels based on the following criteria: high amplitudes with a Z-score >5; a maximum correlation <.4 with any other channel (within a time window of 1 s); predicted channel activity [based on a random 25% of (so far) ‘good’ channels] that has a correlation <.75 with the true channel activity, within a certain fraction (>.4) of non-overlapping 5 s time windows; and unusually high-frequency noise, indicated by the ratio between the median absolute deviations of the high-frequency signal (>50 Hz) versus the low-frequency signal having a Z-score >5 [36]. Residual bad channel detection was run after the entire preprocessing pipeline finished to catch any remaining channels with excessively high or low variance. A participant’s EEG data was also flagged based on a ratio of high-variance time points exceeding .20, and individual channels were flagged based on overall high amplitude. Criteria for excessively high/low variance were voltage thresholds defined as a function of the standard deviation of the voltage measures across all time points or channels, 25 × SD(mV). The criterion for overall high amplitude was a 40-mV threshold for absolute voltage magnitude.”

16. Materials and Methods – Analyses (page 14, lines 269-270): I would encourage to try to explain more how to visually examine exactly for normal distributions.

We have added the following additional information regarding inspection for normality (pg. 16, line 345): “Q-Q plots are presented in Supporting Information (Fig. S1)”.

17. Materials and Methods – Bayesian statistics (page 14, lines 277-278): Please add a reference for this method adopted.

We have added the following reference at the indicated locations (pg. 16, lines 353, 361; and pg. 17, line 370):

van Doorn J, van den Bergh D, Böhm U, Dablander F, Derks K, Draws T, Etz A, Evans NJ, Gronau QF, Haaf JM, Hinne M, Kucharský Š, Ly A, Marsman M, Matzke D, Gupta ARKN, Sarafoglou A, Stefan A, Voelkel JG, Wagenmakers EJ. The JASP guidelines for conducting and reporting a Bayesian analysis. Psychon Bull Rev. 2021 Jun;28(3):813-826. doi: 10.3758/s13423-020-01798-5. PMID: 33037582; PMCID: PMC8219590.

18. Materials and Methods – Bayesian statistics (page 15, lines 283-285): Since Bayesian approach is the key method of this study, it would strengthen this study’s uniqueness if having a detail comparison between Bayesian approach and frequentist approaches in terms of advantages and disadvantages in advance.

We have added the following brief comparison in the first paragraph of the “Bayesian statistics” subsection of the Methods section (pg. 16, lines 357-361): “Bayesian analysis has become increasingly popular in recent years: one of the basic advantages of Bayesian analysis as opposed to frequentist approaches is that it allows researchers to directly quantify the probability of a given hypothesis, rather than calculating the likelihood of observing the data under the assumption of a particular hypothesis; another is that the relative predictive performance of two competing models can be compared [45]”.

19. Materials and Methods – Analysis Plan (page 16, lines 311-313): Please provide a brief result with statistics on collinearity diagnostics.

More explicit results of the collinearity diagnostics are added to the appropriate Results section (pg. 23, lines 482-486): “After running collinearity diagnostics, included confounding variables were sex (Pearson’s r = .066, p = .306) and handedness (r = -.024, p = .710): the planned confounding variables PIQ (r = .319, p < .001, VIF = 1.175), SES (r = .398, p < .001, VIF = 1.226), and age (r = -.126, p = .049, VIF = 1.084) were excluded due to collinearity with PDE (VIF = variance inflation factor)”.

20. Materials and Methods – Analysis Plan: It is encouraged to do some sensitivity analyses to strengthen the robustness of this study’s results.

Thank you for this important suggestion. Sensitivity analyses can be incredibly beneficial in assessing robustness of statistical findings, especially for clinical trial research. Although we have limited experience in conducting sensitivity analyses, we believe that given our data, there are a few options for how we could conduct a sensitivity analysis. For instance, we could (1) repeat our analyses with the inclusion of outliers or (2) without the imputed data for socioeconomic status. Another option is to (3) use varying levels of cutoffs for decoding ability when grouping participants (although this may exacerbate noted disparities in group size for cases where the cutoff is lowered). As our analyses are Bayesian, we could also (4) repeat the analyses using different assumptions for prior distributions. 

As we are less experienced in this arena, we will defer to the reviewer’s recommendation if they had a particular method in mind for this suggestion. However, we hope that Supporting Information Figs S2 and S3 address some of these concerns: Fig. S2 shows robustness checks based on prior specification, similar to option (4), and Fig. S3 shows the accumulated evidence as individual participants are added and the Bayesian model updates. Both allow us to assess the robustness of the reported results, although they may not fit traditional criteria for sensitivity analyses.

21. Discussion (page 21, paragraph 1): It would be beneficial to include brief summarized statements that reinforce the novel contributions of this study to address the existing knowledge gap.

Thank you for your recommendation. We have revised the language in this first paragraph somewhat, and added a brief summary of the more distinguishing aspects of our study on pg. 28, lines 584-592. We changed the word “goal” to “novel contribution”, reiterated that we used “a large, diverse sample” in comparison to previous research, and added the following sentences: “Additionally, we utilized a Bayesian statistical approach to directly quantify the relative likelihood of our hypotheses compared to the null”; and “We hoped to address the current gap in research regarding endogenous phonemic-rate neural oscillations in children with RD compared to typical readers, and in particular hemispheric lateralization of low-gamma rhythms and their relationship to functional differences in this population”.

22. Discussion (where appropriate): It may not be clear what future clinical and practical implications can be derived from this study. Therefore, it would be beneficial to include a paragraph that discusses potential implications.

We added a final section to the Discussion (“Implications and limitations”) addressing the practical implications of our study and its limitations more thoroughly (pg. 35, line 741-): 

“The promise of analyzing endogenous rhythms within the brain is that we will one day be able to reliably identify biomarkers within clinical populations without the need for extensive task-based diagnostic batteries. Additionally, such biomarkers will grant us insight into the mechanisms underlying impairment in reading skills and any shared neural substrates with other well-known comorbid conditions. The current study, although its results are counterintuitive and have very small effects, are consistent with the position that there are possible functional differences between those with and without reading deficits during a passive EEG paradigm. However, any such differences must be met with a considerable degree of caution, given the small effects and the lack of replicability within this field. Given the current state of research, we do not advocate for the use of EEG biomarkers in any clinical setting for the identification of RD or dyslexia.

“One limitation of our study is that in the analyzed sample we had a considerable reduction in perfectly matched populations in favor of a more ‘naturalistic’ approach which allows for the occurrence of comorbid conditions (Table 1). While we tried to correct as much for spurious demographic variables as possible and note that rates of most comorbidities did not differ significantly between the TR and RD groups, there is an implicit degree of uncertainty in how the effects of comorbidities might manifest within each subgroup. This is especially true given our findings in the RD-ADHD subgroup, which were distinct from the RD-only subgroup in relation to temporal lateralization. Finally, it is possible that some of the control group participants may have missed receiving what would have been an appropriate RD diagnosis during childhood but had been able to compensate for any deficits (such that they tested within the normal range on TOWRE) by the time they were assessed. This is a concern that exists throughout reading research, particularly with adults, as it is difficult to objectively measure how much one struggled with reading during childhood retroactively. However, given that the current study was done largely with children under the age of 10, we believe that the chance that a high proportion of the controls had successfully compensated for any previous reading difficulties is relatively low.

“A second limitation are the challenges involved in analyzing resting-state EEG data. There are a few advantages to using resting-state EEG, chief among which are its relative ease of data collection and cross-study comparability relative to task-based designs. Resting-state EEG also has value for characterizing intrinsic functional activity in the brain. At the same time, there are several challenges involved in analyzing and interpreting this data. The first is that in the realm of spectral analysis of EEG data certain frequency bands – such as alpha – are more commonly studied during resting-state, while the gamma band is relatively less well-studied [66]. The consistency of these results may vary as a function of whether the eyes are kept open or closed: an extensive review of spectral analyses for resting-state EEG in various disorders reported that results from EC studies have greater aggregated consistency than those of EO studies [66]. In this study we attempted to elucidate an ‘average’ resting-state gamma profile derived from both EO and EC data, however future studies would do well to examine both states separately with sufficient data for both cases. There is also a degree of ‘fuzziness’ or overlap between functional properties ascribed to these different frequency bands: alpha has been reported to be associated with both performance on various intelligence measures and attention [72, 73], while higher-range gamma activity has also been proposed to be related to attention [64, 67], with alpha-gamma coupling being proposed to reflect early perceptual processing [73-75]. These processes are likely to be relevant not just for attention, but for the processing of visual information such as written language.”

---

## [Decision Letter · Decision Letter 1]

12 Dec 2023

Left-dominance for resting-state temporal low-gamma power in children with impaired word-decoding and without comorbid ADHD

PONE-D-23-28275R1

Dear Dr. Lasnick,

We’re pleased to inform you that your manuscript has been judged scientifically suitable for publication and will be formally accepted for publication once it meets all outstanding technical requirements.

Kind regards,

Thiago P. Fernandes, PhD

Academic Editor

PLOS ONE

Additional Editor Comments (optional):

Thank you for your thoughtful edits; commendable for the refinement and endeavour.

By my own reading, the ms now reads better, and the authors addressed the concerns. 

Wishing you success with the study.

Reviewers' comments:

Reviewer's Responses to Questions

**Comments to the Author**

1. If the authors have adequately addressed your comments raised in a previous round of review and you feel that this manuscript is now acceptable for publication, you may indicate that here to bypass the “Comments to the Author” section, enter your conflict of interest statement in the “Confidential to Editor” section, and submit your "Accept" recommendation.

Reviewer #2: All comments have been addressed

2. Is the manuscript technically sound, and do the data support the conclusions?

Reviewer #2: Yes

3. Has the statistical analysis been performed appropriately and rigorously? 

Reviewer #2: Yes

4. Have the authors made all data underlying the findings in their manuscript fully available?

Reviewer #2: Yes

5. Is the manuscript presented in an intelligible fashion and written in standard English?

Reviewer #2: Yes

6. Review Comments to the Author

Reviewer #2: The authors have carefully and diligently addressed all my comments. Thanks for the great efforts and I have no further suggestions.

7. PLOS authors have the option to publish the peer review history of their article (what does this mean?). If published, this will include your full peer review and any attached files.

Reviewer #2: No

---

## [Editor Report · Acceptance letter]

18 Dec 2023

PONE-D-23-28275R1 

PLOS ONE

Dear Dr. Lasnick, 

I'm pleased to inform you that your manuscript has been deemed suitable for publication in PLOS ONE. Congratulations! Your manuscript is now being handed over to our production team.

Kind regards, 

on behalf of

Dr. Thiago P. Fernandes 

Academic Editor

PLOS ONE